# Comparison of the Capacity of Several Machine Learning Tools to Assist Immunofluorescence-Based Detection of Anti-Neutrophil Cytoplasmic Antibodies

**DOI:** 10.3390/ijms25063270

**Published:** 2024-03-13

**Authors:** Daniel Bertin, Pierre Bongrand, Nathalie Bardin

**Affiliations:** 1Service d’Immunologie, Biogénopôle, Hôpital de la Timone, Assistance Publique-Hôpitaux de Marseille (AP-HM), 13005 Marseille, France; daniel.bertin@ap-hm.fr (D.B.); nathalie.bardin@ap-hm.fr (N.B.); 2Laboratoire Adhésion et Inflammation, Aix-Marseille University, UM61, Campus Luminy, 13009 Marseille, France; 3INSERM U1067, Campus Luminy, 13009 Marseille, France; 4CNRS, U7333, Campus Luminy, 13009 Marseille, France; 5INSERM, U1076, Campus Timone, 13005 Marseille, France; 6Aix Marseille University, INSERM, INRAE, C2VN, Campus Timone, 13005 Marseille, France

**Keywords:** artificial intelligence, ANCA, immunofluorescence, vasculitis, image analysis, myeloperoxdase, proteinase 3

## Abstract

The success of artificial intelligence and machine learning is an incentive to develop new algorithms to increase the rapidity and reliability of medical diagnosis. Here we compared different strategies aimed at processing microscope images used to detect anti-neutrophil cytoplasmic antibodies, an important vasculitis marker: (i) basic classifier methods (logistic regression, k-nearest neighbors and decision tree) were used to process custom-made indices derived from immunofluorescence images yielded by 137 sera. (ii) These methods were combined with dimensional reduction to analyze 1733 individual cell images. (iii) More complex models based on neural networks were used to analyze the same dataset. The efficiency of discriminating between positive and negative samples and different fluorescence patterns was quantified with Rand-type accuracy index, kappa index and ROC curve. It is concluded that basic models trained on a limited dataset allowed for positive/negative discrimination with an efficiency comparable to that obtained by conventional analysis performed by humans (0.84 kappa score). More extensive datasets and more sophisticated models may be required for efficient discrimination between fluorescence patterns generated by different auto-antibody species.

## 1. Introduction

*Potential of artificial intelligence to improve medical practice*. The steady growth of the diversity, power and cost of therapeutic tools is an incentive to attempt to increase the precision of diagnosis without a parallel increase of expenses. The spectacular progress of computer-based methods, referred to as artificial intelligence (AI) or machine learning (ML), may be of considerable help in this respect by allowing for the extraction of maximal information from biological data with optimal rapidity and minimal recourse to biological experts. Despite initial disappointment several decades ago along this line [1], the tremendous progress of machine learning algorithms has resulted in a steady development of the use of AI in medicine [2], whether to process large datasets generated by multi-omic methods in order to elaborate general prediction algorithms [3], to identify new markers of clinical interest [4] or to analyze the output of standard biological tests performed on individual patients in order to achieve more rapid, more reliable and less costly diagnoses [5].

*A representative example: use of fluorescence microscopy for autoimmunity diagnosis*. Indirect immunofluorescence has long been considered an important tool, and even a so-called gold standard, to detect anti-nuclear antibodies (ANAs) associated to severe conditions such as systemic lupus erythematosus [6,7], or anti-neutrophil cytoplasmic antibodies (ANCAs), which are associated with a number of vasculitis-involving syndromes [8]. The basic principle consists of exposing fixed cells to patients’ sera and looking for the presence of auto-antibodies by microscopical observation of slides labelled with fluorescent anti-immunoglobulin antibodies. ANAs are usually detected on Hep-2 cells, and ANCAs on polymorphonuclear leukocytes. An experienced pathologist is required to recognize specific patterns revealing the presence of suspected antibodies. Thus, the examination of fluorescence patterns on ethanol-fixed leukocytes may reveal so-called cellular-type ANCAs (C-ANCAs), with a cytoplasmic pattern usually associated with anti-proteinase 3 antibodies, or perinuclear-type ANCAs (P-ANCAs), usually associated with anti-myeloperoxidase antibodies [9]. Other patterns may be due to ANAs or antibodies of other specificities that may be indicative of different pathological situations with different therapeutic implications [10,11]. The well-recognized finding [12,13,14] that inconsistencies may occur between different laboratories is a strong incentive to attempt to standardize the processing of immunofluorescence images [15].

*Computer-assisted image classification*. Two main strategies may be considered. The earliest approach consisted of defining, measuring and processing numerous so-called hand-crafted texture parameters [16,17]. Classification could then be performed by processing these parameters, also called attributes or features, with well-established ML algorithms such as k-nearest neighbors (KNN), support vector machines (SVM) or random forest [18]. Classification efficiency was essentially dependent on the quality of image parameters [19], and reported models might include more than one thousand parameters [16] and make use of advanced mathematical concepts [20]. More recently, the outstanding progress of AI has made it an attractive prospect to use more potent algorithms, such as neural networks, to perform both feature extraction and classification. This so-called deep learning approach has indeed met with impressive success in important domains such as text or facial recognition [21,22]. Further, the availability of these algorithms is strongly increased by the development of open access platforms, such as scikit-learn (http://scikit-learn.org, accessed on 10 March 2022) or tensor flow (https://www.tensorflow.org/, accessed on 15 October 2023), that include exhaustive online documentation and the use of which is facilitated by excellent written tutorials [23,24,25]. Accordingly, these platforms are currently used in state-of-the-art research projects [26,27].

*Quantification of the efficiency of image classification*. An essential requirement to foster progress is to make use of objective tools for measuring the efficiency of different classification methods. The fraction of accurate predictions, which may be designated “predictive accuracy” (*pa*), is a widely used and fairly intuitive reporter of the efficiency of binary classification [12]. However, it may provide a less appropriate measure of the efficiency of multiclass data partition, which may be more precisely represented by parameters such as Rand index [28]. Also, a calculated classification accuracy may be deceptive. Indeed, if an algorithm is used to detect positive samples in a batch of sera that are mostly negative—a quite common situation—a very high accuracy may be obtained by classifying all samples as negative! A widely used correction [29] consists of calculating the accuracy increase provided by a model as compared to random agreement following a simple equation, yielding the so-called Cohen *kappa index*.
*ka* = (*pa* − *random pa*)/(*maximum pa* − *random pa*) (1)
where *ka* is for kappa index, *random pa* is the precision accuracy corresponding to a random choice, and *maximum pa* is the precision provided by a fully exact model. This index has been used in numerous reports of diagnostic accuracy [13,30,31]. It was suggested that that the agreement be considered either moderate, substantial or perfect when *kappa index* is, respectively, higher than 0.4, 0.6 or 0.8 [29]. It must be kept in mind that this index is dependent on diagnostic criteria as well as the specific features of the sample population used to perform the comparison between a given model and the gold standard. It may thus be appropriate to mention “kappa-type measures” [29] and give details on the exact algorithm used to calculate *kappa index*. Further, a more exhaustive account of the efficiency of a method is provided by plotting *sensitivity* (i.e., the fraction of positive samples that are classified as positive) versus *one minus specificity* (where specificity is the fraction of negative samples that are classified as negative). This is called the receiver–operator-characteristic (ROC) curve [32], and the model efficiency is expressed as the area under the curve (*auc*) that is expected to be between 0.5 (corresponding to a random classification) and 1 (corresponding to a perfect classification). It is important to recall that this curve is dependent on the population used to perform the comparison. In a recent meta-analysis of 56 reports [33], the minimal *auc* value required for a test to be considered good or very good ranged between 0.75 and about 0.95. Also, it may be useful to use an index suited to both binary and multilabel classification. A Rand-type index corrected for random agreement elaborated by the scikit-learn team was found convenient in a recent study [34]. This index will be designated here as corrected predictive accuracy (*cpa*). As shown below, this was found to be tightly related to *pa*, kappa index and *auc* (see Section 4.1).

*Current status of automated microscopic-based autoimmunity diagnosis*. The recourse to human analysis for ANA or ANCA detection is well known to raise several difficulties, such as discrepancies between different readers or difficulty with standardizing tests for certification of medical laboratories [15,35]. Automatic methods based on hand-crafted parameters have long been elaborated to address this problem [16,36]. This allows for the development of commercially available systems that reliably perform simple tasks such as discrimination between positive and negative samples [37]. Thus, a simple algorithm elaborated in our laboratory allowed for safe discrimination between ANA-positive and -negative samples with a kappa coefficient of 0.92 [31], and a comparison of 6 commercial systems yielded an *auc* of order 0.95 for this task [37]. However, the recognition of specific patterns appeared more challenging, with a recognition efficiency of commercial systems varying between about 40% and 85% [37]. More recently, a combination of numerical feature engineering and ML allowed ANAs to be partitioned into 5 classes with an *auc* of 0.95 [20]. Positive/negative discrimination of ANCAs was achieved by a commercial system with *pa* of 0.86 [13]. However, deep learning based methods displayed impressive progress and, as recently reviewed [12], numerous reports have described the application of neural networks to the recognition of ANA patterns, and these have been recently found to outperform traditional methods. Thus, traditional texture-based methods were found to yield 0.93 *pa* as compared to 0.95 found with InceptionResnetV2, a recently described convolutional network [18,38]. In another study, the capacity of 6 public convolutional networks to classify ANAs was studied: comparison with experienced readers yielded kappa indices between 0.63 and 0.82 (1985 samples). Interestingly, kappa index was only 0.528 when experienced and beginner users were compared [17].

However, ANCA detection may be considered more demanding, since neutrophils display more complex nucleus shapes than Hep-2, which is used for ANA detection [30,35]. Also, while immunofluorescence studies performed for ANA detection are carried out with the Hep-2 cell line, neutrophils used for ANCA detection may display substantial heterogeneity (see Section 4.2).

*Purpose of the present report*. Our aim was to present a detailed description of the potential of basic machine learning tools to classify immunofluorescence images used for ANCA detection. We first built a dataset including 1733 cell images obtained by processing 137 sera. Two strategies were followed. First, microscopic images were used to extract four features suggested by biological experience and assess the classification efficiency of simpler machine learning tools including logistic regression, k nearest neighbors classifier and decision tree. Secondly, cell images (50 × 50 pixels) were subjected to individual analysis with aforementioned models and more complex neural networks. It was concluded that *kappa scores* higher than 0.8 could be obtained for discrimination between positive and negative samples with a limited dataset and fairly simple tools. However, our preliminary attempts suggest that achieving efficient pattern classification will require a more extensive dataset in order to select and to train more recent and sophisticated models. Currently available and promising strategies are described in the discussion.

## 2. Results

Representative cell images are shown in Figure 1.

### 2.1. Combination of Biologically Inspired Indices and Machine Learning

Data were first used to perform a binary classification between ANCA-positive and -negative samples. Secondly, we tried to discriminate between immunofluorescence patterns.

#### 2.1.1. Discrimination between Positive and Negative Samples with Full Image-Related Indices

First, full images (encompassing entire microscopic fields) generated with individual sera were processed to derive four quantitative indices that were felt to have a possible relevance to ANCAs, as explained in Section 4.4.1. This yielded a dataset comprising 137 samples that were classified as negative (102/137) or positive (35/137) by conventional analysis. This dataset was randomly split 100 times into a training set (102 samples) and a testing set (35 samples). As suggested by a previous comparison of the efficiency of eight standard algorithms used to analyze limited datasets [34], we selected three fairly simple algorithms: logistic regression (LR), k-nearest neighbors (KNN) and decision trees (DT). The prediction accuracy (*pa*), corrected prediction accuracy (*cpa*) and area under ROC curve (*auc*) obtained on training and testing sets after training on training sets are shown in Table 1.

Representative ROC curves are shown in Figure 2.

While classification efficiency might be considered fairly good as compared to other studies, efficiency parameters were significantly lower than one. Also, LR significantly outperformed KNN and DT (*p* = 0.00013). It was important to explore different means of improving this situation.

We tried a simple neural network (multilayer perceptron) as a more elaborate model: classification efficiency was not improved (testing cpa = 0.59, test *auc* = 0.83), in accordance with our earlier conclusion that simpler ML models were better suited to processing limited datasets [34].Since ML is considered fairly “data hungry” [39], it was of interest to ask whether an insufficient dataset size (137 samples) might be an important cause of prediction errors. This question was addressed by measuring the dependence of LR classification efficiency on sample number. As shown in Figure 3, *index-based* classification efficiency was only weakly dependent on the dataset size.The behavior of ML algorithms is dependent on so-called hyperparameters that are often ignored, since default values are usually satisfactory. It was checked that the classification efficiency of LR could not be improved by changing LR regularization parameter C. As expected, the default value (C = 1) was found satisfactory. Reducing regularization resulted in significant increase of training *cpa*, with a decrease of testing cpa, which was indicative of overfitting. Increasing regularization resulted in concomitant decrease of *cpa* on training and testing datasets.Aforementioned results strongly suggested that classification efficiency was limited by the intrinsic capacity of indices used to quantify images, in line with conventional wisdom [19]. Since the first index was derived from our experience of automatic detection of anti-nuclear antibodies [15,31], we tested the discrimination provided by this sole index, based on empirical determination of a threshold value separating positive from negative samples. Our dataset was randomly split 100 times between a training set (102 samples) and a testing set (35 samples). The average *cpa* parameters obtained on the training and testing sets were, respectively, 0.705 +/− 0.004 SE and 0.701 +/− 0.013 SE, which were slightly but significantly (*p* = 0.0016) higher than efficiency parameters shown in Table 1. This supports the well-known fact that addition of improper features may hamper LR efficiency.

#### 2.1.2. Automatic Discrimination between Several Fluorescence Patterns

It was felt to be of interest to determine whether ML could help us discriminate between different fluorescence patterns. We investigated the possibility of automatic discrimination between the 9 C-ANCA-positive and 26 P-ANCA- or ANA-positive images yielded by the 35 positive sera included in our dataset. Interestingly, a preliminary study revealed that none of the four aforementioned indices individually allowed for any discrimination between both groups: indeed, the efficiency parameter *cpa* obtained by separating both groups was, respectively, 0.0447 +/− 0.1530 SD, 0.0545 +/− 0.2101 SD, 0.0300 +/− 0.2127 SD and 0.0252 +/− 0.1542 SD with indices i1 to i4. However, when the four-parameter dataset was processed with three ML algorithms, a poor but significant discrimination between C-ANCA and P-ANCA was obtained. LR and KNN displayed comparable *cpa* and significantly (*p* = 0.00027) outperformed DT. Results are displayed in Table 2.

Finally, ML was used to process the whole 137 sample dataset in order to try and discriminate between four groups of interest: negative (102/137), C-ANCA (9/137), P-ANCA (21/137) or atypical patterns due to ANAs (5/137). As shown in Table 3, a substantial discrimination was observed, and LR significantly outperformed KNN (*p* = 0.007) and DT (*p* = 024) as found by comparing testing cpa.

The relatively small difference between performance parameters obtained with three different algorithms and impossibility to improve agreement with hyperparameter adaptation suggested that the limitation was essentially due to an insufficient discriminative power of the four-feature description used to account for image properties. However, these results were consistent with the widespread hypothesis that a simple strategy for achieving automatic classification of ANCA-related images might consist of adding an increasing number of texture parameters and automatically combining them with simple ML algorithms. Indeed, ANA pattern analysis was performed with a commercial system involving 1400 object-describing parameters [40]. However, recent progress of AI was an incentive to look for a fully autonomous way of analyzing immunofluorescence images. Results obtained along this line are shown below.

### 2.2. Use of AI for Autonomous Analysis of Fluorescence Images

Two strategies were considered: (i) combining data reduction with fairly simple ML algorithms. (ii) using neural networks for complete analysis.

#### 2.2.1. Use of Data Reduction to Process Individual Cell Images

The description of analyzed microscope fields with only four global parameters was replaced with the use of a 2500-parameter set (50 × 50 pixel intensities) to account for each cell image contained in a given microscope field. Fifty-one sera were used to build a dataset of 1733 individual cell images (513 negative, 309 C-ANCA, 789 P-ANCA and 122 atypical patterns that could be ascribed to ANAs).

In the first step, the capacity of aforementioned three standard ML algorithms to analyze these images without any data reduction was studied. As shown in Table 4, parameter *cpa* obtained for positive/negative discrimination was fairly low. Since the important difference between training and testing *cpa* was indicative of overfitting that might be ascribed to an excessive number of features as compared to the sample number, we used principal component analysis as a standard way of reducing the number of parameters. As shown in Table 4, this resulted in a significant increase of efficiency parameters, since the highest testing *auc* was raised from 0.86 to 0.92, and the highest testing *cpa* was increased from 0.35 to 0.46 (*p* < 10^−5^). Representative ROC curves are shown in Figure 4.

The possibility of discriminating between nuclear/perinuclear and cytoplasmic fluorescence was also studied. As shown in Table 5, standard algorithms displayed a poor capacity to discriminate between both patterns, since the maximum *cpa* value was 0.11 when the algorithms were used on full images or on the first 20 principal components. KNN significantly outperformed LR and DT (*p* < 10^−5^).

Since images were expected to include the information required to discriminate between C-ANCA and P-ANCA, it was of obvious interest to try and determine why standard ML algorithms were unable to select the relevant information. A likely possibility might be that each serum generated a particular fluorescence pattern in addition to a “general” cellular or nuclear localization. Training would thus result in a capacity of an ML model to recognize patterns specific to the particular antibody set of each tested serum. This possibility was addressed by visualization of the first two principal components of images displayed by six sera (3 cANCA, 3 pANCA), as shown in Figure 5. A clearcut separation could be observed between images generated by different sera of similar (either C-ANCA or P-ANCA) specificity.

This supported the need for a more refined ML algorithm allowing for precise selection of desired features. In view of the remarkable success obtained with neural networks in the field of image analysis, it was deemed appropriate to use a number of neural networks to analyze our dataset.

#### 2.2.2. Analysis of Full Images with Neural Networks

While recent successes met by neural networks in the field of image analysis were an incentive to explore the potential of this model class, a major problem is that a neural network may involve a high number of hyperparameters. First, we used multilayer perceptron as relatively simple models, and the importance of three major hyperparameters (hidden layer number, hidden layer size and regularization parameter) is shown in Figure 6.

The following conclusions were suggested:Efficiency parameters displayed limited change in response to fairly extensive variation of hyperparameters, suggesting a moderate dependence of classification efficiency on the model settings.Parameter *cpa* calculated on testing sets varied between a minimum value of 0.38 and a maximum of 0.51 (with *kappa score* and *auc*, respectively, equal to 0.67 and 0.85). Neural network performance was thus better than that achieved with standard ML models (shown in Table 4).Plots displayed in Figure 6C,D clearly confirmed the risk of overfitting as a consequence of insufficient regularization (C) or excessive number of features (D) as compared to the number of samples, leading to a high *cpa* training/*cpa* testing ratio.

These results were an incentive to investigate the capacity of MLP to discriminate between different fluorescent patterns. When a dataset of 1220 images with cellular (309) or nuclear/perinuclear (911) fluorescence localization was studied with a wide range of hyperparameter settings, the testing *cpa* ranged between 0 and 0.11, suggesting that this dataset was insufficient to allow for proper model training, as was also found with simpler models (Table 5).

An important point is that high *cpa* values for positive/negative and pattern discrimination of the images of *training sets* were obtained, since the highest values were, respectively, 0.99 and 0.92. These results suggested the conclusion that ML models were sufficiently versatile to efficiently discriminate between all images, but the classification criteria obtained with autonomous training did not match the biologically significant classification, resulting in marked overfitting.

It was important to know whether the fairly modest classification efficiencies displayed in Table 4 and Table 5 were due to an insufficient size of our dataset. In order to address this point, we studied the sample number dependence of the classification efficiency of KNN and MLP. It was considered interesting to know whether our dataset displayed a specific behavior when compared to others. Therefore, we subjected another dataset to the same study: we used the public fashion-MNIST database [41], which includes 10 series of 28 × 28 pixel images representing cloth styles. Two classes were selected (T-shirts and trousers), and varying numbers of random combinations of these two types were processed for binary classification. Results are shown in Figure 7, suggesting the following conclusions: (i) increasing the sample number resulted in a steady increase of testing cpa under all studied conditions. (ii) Cell image classification was much less efficient than that of the fashion-MNIST images, suggesting the need for much higher sample numbers. (iii) A remarkable behavior was displayed by MLP classification of ANCA images subjected to dimensional reduction: while testing cpa displayed steady increase when sample number was increased, train *scpa* first decreased then decreased (Figure 7B), illustrating the aforementioned possibility that individual samples could be identified on features unrelated to pos/neg classification when the sample number was low.

A possible reason for the limitation of MLP efficiency is that the localization of individual pixels might not be sufficiently apparent in parameter sets organized as 1-dimensional arrays. Indeed, convolutional neural networks may be considered better suited than networks including only so-called dense layers to detect specific image patterns when they are trained with 2-dimensional arrays. Thus, we tentatively assayed the capacity of a series of convolutional neural networks to perform positive/negative classification of 1733 images datasets. Test *cpa* values of 0.52, comparable with those obtained with MLP, were obtained. Also, no significant improvement of pattern classification was obtained with the aforementioned 1200 image dataset.

A possible explanation for neural network limitation might be that differences between individual sera and cells might generate fluorescence variations overlapping with the effects of antibody specificity. A simple way of testing this possibility consisted of performing controlled splitting of image sets by ensuring that all images generated by a given serum were in the same (training or testing) set. Results obtained with this strategy are shown below.

#### 2.2.3. Combination of Controlled Splitting of Training and Testing Datasets and Serum Rather than Image Classification

The following two modifications of processing were performed: (i) it was ensured that all images generated from the same serum fell into the same (training or testing) dataset; (ii) after training models on 2500-pixel images as indicated above, sera belonging to testing datasets were classified as positive or negative according to the highest number of individual cell images classified as positive or negative.

As shown in Table 6, controlled splitting did not improve individual cell classification, but the modified procedure resulted in highly significant improvement of serum classification, since testing *cpa* was 0.74 with MLP (the corresponding kappa score was 0.84).

It was investigated whether this controlled splitting might improve pattern classification. However, classification efficiency was not improved—and was in fact significantly decreased—when this controlled splitting was performed.

## 3. Discussion

The main purpose of this work was to investigate a fairly simple and well-defined problem of current medical interest to explore the possibility of autonomous building of ML models with sufficient reliability to assist and possibly replace biological experts in analyzing microscopic images. It was hoped that this endeavor might help in identifying general guidelines, limitations and possible strategies for future progress.

*A first conclusion* is that currently available ML algorithms autonomously trained on a fairly restricted dataset were able to perform positive/negative discrimination matching experienced human analysis, with a kappa score of 0.84, which is considered very good [17,29]. This simple finding is of actual clinical interest: first, automatic positive/negative discrimination would be most useful by decreasing current delays in delivering a negative diagnosis or performing ELISA assays to check positivity. Secondly, experienced readers could spend more time studying positive samples if they were not asked to analyze negative ones that usually represent the majority of tested sera.

It must be emphasized that the estimated values of achieved efficiency parameters may be considered fairly reliable despite the lack of a fully independent validation dataset. Indeed, average values of efficiency indices were estimated after repeated—up to 100-fold—splitting of the full dataset, and results shown in Figure 5 and other tests performed on basic classifiers suggest that these efficiency parameters were not strongly dependent on model hyperparameters, thus making less likely the possibility that calculated efficiency indices might be artefactually high due to a strong influence of the dataset on parameter choice. However, as was recently emphasized, there is a need to perform additional validation tests. Indeed, patients’ demographic properties might exhibit temporal or local [42,43] changes, thereby hampering model validity through a so-called dataset shift. Thus, the validity of our results should be checked by studying sera tested in our laboratory during different periods of time and by organizing multicenter trial.

*The second conclusion* is that our simple approach did not succeed in safely discriminating between different fluorescence patterns. Many non-exclusive strategies may be considered to improve this situation:

*Quantitative and qualitative dataset improvement*.
Our results (e.g., Figure 7) strongly suggest that a *quantitative expansion* of our data base is needed to achieve reliable pattern classification. This is in line with the common opinion that the success of ML is largely due to the mining of large amounts of data [19,25] (p. 27) [44].The information provided by individual serum samples might be enhanced by performing additional fixation or staining procedures. Indeed, it has long been reported that the localization of ANCA-related fluorescence is not the same on ethanol- and paraformaldehyde-fixed cells [9], and it might be more informative to use datasets including two fluorescence images. Also, nuclear localization provided by DAPI labelling could also be inserted in an additional channel. CNNs would be well suited to analyzing image stacks associated to individual cells, and DAPI staining was used in recent attempts at ANA classification with ML [17]. Also, it might not be warranted to increase the complexity and cost of immunofluorescence testing if this did not result in a very substantial increase of information content.Different *image preprocessing* procedures might be considered, such as filtering to remove noise or replacing image resizing by embedding into larger areas to retain information relevant to absolute distances.A common way of increasing ML power consists of increasing feature diversity [45,46]. Thus, it might be rewarding to combine images with other patients’ features. However, the need for additional parameters that might not be immediately available in hospital laboratories would delay computer-assisted analysis, thus hampering ML-generated rapidity gain. Therefore, specific clinical trials would be needed to validate feature extension.Training a model with a restricted dataset might be improved with *data augmentation*, which consists of creating “realistic” data with suitable algorithms. As an example, the classification of macrophages from microscopic images with simple geometrical features such as area or circularity was reported to display an accuracy increase from 0.3 to 0.93 when the dataset size was increased one hundred-fold with a custom-made image generator [47].

### 3.1. Model Choice

As indicated in the introduction, two main strategies may be considered when analyzing a given dataset. Processing hand-crafted features with a ML model, or performing both feature extraction and classification with ML.

As shown in the first part of this report (Table 1), the use of biologically inspired indices is an attractive way of combining biological expertise and AI. Indeed, many commercially available systems successfully use ML algorithms to process extensive sets of texture parameters. However, the development and continuous improvement of an algorithm involving more than 1000 parameters [40] may be more difficult to perform than the autonomous building of ML models. Accordingly, recent comparisons between deep learning and a combination of hand-crafted features and simple ML models such as SVM or random forests supported the superiority of neural networks [18,20]. However, it would be an attractive prospect to use ML to improve the power of selected parameters. While neural networks are often compared to “black boxes”, theoretical effort is currently underway regarding “interpreting” their behavior [48]. These endeavors might in the future help improve biological intuition and thereby allow for substantial improvement of so-called hand-crafted features.Results presented in this report revealed a significant but insufficient efficiency of a combination of data reduction with PCA and simple ML methods to classify 50 × 50 pixel images. Indeed, neural networks may now be considered the gold standard for image analysis [12], and they are currently the basis of many current reports on ML classification for medical purposes [49]. However, while more and more powerful network architectures are continually being reported and tested [17,18,50], model setting and training quality are essential determinants of final performance. Available strategies will be rapidly listed below.

### 3.2. Hyperparameter Setting

As exemplified in this report, the performance of a given ML model is dependent on the choice of hyperparameters. Unfortunately, there is currently no general tool allowing for the prediction of the best set of models and hyperparameters to solve a given problem, and the search for such tools is indeed an active field of study [51]. Thus, *an empirical search* is needed. This may be performed either by systematic testing on a grid (Figure 7) or by random attempts. However, an exhaustive study may not be feasible if hundreds of combinations have to be tested and the training time is fairly high. As an example, several tens of hours have been reported to be needed to train complex CNNs, even when a graphic card was used to increase computing speed [17].

However, hyperparameter choice may be facilitated by some empirical rules, such as the setting of regularization parameters to provide sufficient versatility to fit training data without generating overfitting.

### 3.3. Improving the Training Process

A common means of reducing overfitting consists of stopping the training phase as soon as the validation error reaches a minimum. While this *early stopping* procedure is widely used and intuitively considered reasonable, the identification of an optimal training duration may warrant further studies [52,53].Another procedure facilitating the training of very complex models, dubbed *transfer learning*, consists of using a pretrained model and training only the outer layers to fit a specific dataset. This method permits the use of highly successful models trained on public image datasets such as ImageNet [54] with a reasonable computing load for ANA classification with Hep-2 cells [50,55].An attractive prospect might consist of driving the development of a complex model through what might be dubbed *smart training* and yield unexpected performance. Thus, the development of a convolutional structure in a fully connected network was achieved by training this network with translation-invariant data [56]. Also, a neural network was claimed to acquire increased capacity through a special learning method dubbed meta-learning [57].

Thus, recent successes have led to the conclusion that modern deep neural networks for image classification have achieved superhuman performance [53], leading to the conclusion that the failure in dealing with a given image analysis classification problem might result from the incapacity to test a sufficient fraction of the overwhelming amount of models and setting procedures currently available.

An important point that does not fit into the scope of the present study is that in the near future, AI alone or in combination with human expertise should improve the reliability of diagnosis and medical decision. This might result in an improvement of medical practice, provided future clinical trials are performed to validate better “gold standards”. However, due to the possibility of checking immunofluorescence data with ELISA, the main interest of ML should first be restricted to a gain of rapidity.

## 4. Materials and Methods

### 4.1. Patients

This retrospective study was performed on 137 sera sent by clinical departments and processed in the immunology laboratory of Marseilles public hospitals for the detection of antineutrophil cytoplasmic antibodies (ANCAs). All samples received within a fixed period of time were kept without any selection. Serum samples were part of the Marseilles Biobank (registered as DC 2012_1704), and the study was approved by the medical evaluation board and health data committee of Assistance Publique-Hôpitaux de Marseille, Marseille, France and fulfilled local requirements in terms of data collection and protection (GDPR 2019-133). The laboratory is accredited by the French Cofrac for immunological tests (certificate 8-1739).

### 4.2. Immunofluorescence

ANCAs and their staining patterns (perinuclear, cytoplasmic) were detected by IF on ethanol-fixed human neutrophil slides (Immuno Concepts, Sacramento, CA, USA) according to the supplier’s recommendations, except for the following extended labeling protocol: Serum samples diluted in phosphate-buffered saline were added for 30 min at room temperature (RT). After washing, bound antibodies were labelled by incubation with fluorescein isothiocyanate (FITC)-conjugated sheep anti-human immunoglobulin (Immuno Concepts, CA, USA) for 30 min at RT. Slides were then washed and embedded with a 4,6-diaminophenylindol (DAPI)-containing medium (Vectashield, Vector laboratories Inc., Burlingame, CA, USA) for nuclear staining.

For each patient, two images of the same central microscopic field were automatically captured with 20× objective at two different excitation wavelengths: 480 nm for FITC stain and 360 nm for DAPI stain. We used a fully robotized fluorescence microscope (Axio Imager M2, Carl Zeiss, Jena, Germany) equipped with an automated 200-slide handling system (SlideExpress, Märzhäuser, Wetzlar, Germany) and with 360-nm and 480-nm LEDs for excitation (Colibri-2 LED illumination system, Carl Zeiss, Jena Germany). Images with 1360 × 1024 pixel resolution were captured using a monochrome CCD camera (ProgRes® MF Cool camera, Jenoptik, Germany) with a pixel size of 6.45 μm ^2^. Exposure times for FITC and DAPI captures were 70 ms and 200 ms, respectively. All captured grayscale images had an 8-bit-depth and were saved in tagged image file format (TIFF) as previously described [15].

Images were examined by an experienced biologist and ELISA testing was performed for diagnosis confirmation when they were classified as positive.

### 4.3. ELISA Testing

Following standard practice [8], the specificity of samples classified as positive was checked by looking for anti-myeloperoxidase or anti-proteinase-3 antibodies via enzyme-linked immunoassay (ELISA, Euroimmune, Lübeck, Germany), according to the supplier’s specifications.

### 4.4. Image Processing

#### 4.4.1. Calculation of Overall Quantitative Indices

Images were first processed with a previously described system performing automated image recording and positive/negative ANA classification (ICARE) [31] which was written in Java as a plugin to Image J V1.53 [58]. ICARE was supplemented with a specific custom-made plugin for automatic determination of four quantitative indices that were felt relevant to the presence of ANCAs. Briefly, the DAPI image was used to define cell surface as the set of pixels with an intensity at least four times higher than the first peak on the intensity histogram. Unexpectedly, while this matched the nucleus on Hep-2 cells (used for ANA detection), it filled whole neutrophil surfaces, likely due to the contorted nucleus shape and possibly particular cytoplasmic staining properties or this cell population. This was used as a basis for the determination of the following four indices:-*Index i1* is the ratio between the mean intensity on FITC images of pixels classified as “inside” and “outside”. This was expected to permit discrimination between positive and negative samples.-*Index i2* is the ratio between the mean FITC intensity of pixels defined as “inside” and the first peak intensity of the histogram of FITC image.-*Index i3* is similar to i2, but “inside” is defined on DAPI histograms as pixels with an intensity 16 times higher than that of the first background peak. It was expected that this region might be closer to actual nuclear regions.-*Index i4* is the correlation between FITC and DAPI pixel intensities in regions defined as “inside” on DAPI images. It might be hoped that the correlation would be highest with ANA, lowest with C-ANCA and intermediate with P-ANCA.

Images were simultaneously classified by an experienced pathologist as negative (0), C-ANCA (1), P-ANCA (2) or atypical/ANA type (3) and processed by ICARE [31]. A file including 137 samples with 4 indices each was thus prepared for ML processing. This provided a dataset consisting of 137 samples (102 negative, 9 C-ANCA, 21 P-ANCA, 5 atypical/ANA).

#### 4.4.2. Building Individual Cell Images

Individual cell images were built out of whole microscopic fields according to the following two-step procedure:

First, cell boundaries were determined on DAPI images with a threshold-based algorithm that has been used for decades in our laboratory [59] and used as a Java plugin for Image J. The same threshold (60/256) was found to be convenient for all 137 images.

Secondly, rectangular areas enclosing cell boundaries were resized to 2500 = 50 × 50-pixel images by plain homothety with a custom-made Python program and stored as CSV files for further treatment. This allowed for the building of a dataset of 1733 individual cell images (513 negative, 309 C-ANCA, 789 P-ANCA, 122 atypical/ANA) out of 51 sera, including all 35 positive sera and 16 negative sera randomly selected from the 102 negative samples.

### 4.5. Machine Learning

#### 4.5.1. Classification Based on “Hand-Crafted” Parameters

The 137-sample CSV file was processed as a four-parameter NumPy array with four standard models provided by scikit-learn. Default hyperparameters were retained unless otherwise mentioned.

Three conceptually simple models were found most efficient to process limited samples involving few features [34]: *logistic regression classifier (LR)* is inspired from conventional multivariate statistics and is based on the tentative estimate of posterior probabilities of several classes with linear functions [60]. The *k nearest neighbor* (KNN) *classifier* will partition a given sample according to the majority of its k nearest neighbors in the feature multidimensional space. K may be modified at will by changing *n_neighbors* hyperparameter. Importantly, the choice is dependent on the definition of distance in this space, which may be modified by scaling. The *decision tree* (DT) *classifier* will partition a sample by performing a series of binary tests, the total number of which may be limited by *max_depth* hyperparameter.

*Multilayer perceptron* (MLP) is a simple neural network. This is made of a series of sequential *layers* including so-called *nodes* or *neurons* that send output signals to the following layer as functions of a combination of input signals sent by the underlying layer. Neural networks involve a very high number of parameters, since the interaction between each couple of nodes belonging to adjacent layers is characterized by a so-called weight parameter that is set during the training phase. This complexity results in a high capacity to fit extended datasets. The denomination of “deep learning” refers to the involvement of a possibly high number of layers inserted between the “external” input and output layers. The behavior of these so-called hidden layers is usually largely ignored.

Since the dataset was not extensive enough to allow us to optimize hyperparameters (i.e., fixed, training-independent parameters), we essentially used default values with minimal changes that were found suitable for a low feature number dataset (this was 4 as indicated above) [34]. For each method, the dataset was split 100 times into a training set and a testing set, and classification efficiency was obtained by calculating prediction accuracy (*pa*), corrected prediction accuracy (*cpa*, a modified Rand-type score corrected for chance, as calculated with scikit-learn adjusted_rand_score function), Cohen kappa score and area under ROC score (*auc*) when positive/negative discrimination was studied.

#### 4.5.2. Analysis of Individual Cell Images

Individual cell images (50 × 50 pixels) were first subjected to a scaling procedure (scikit-learn RobustScaler method) to ensure that all parameters displayed similar median and quartile distributions. In some cases, data reduction was performed with principal component analysis (PCA).Images were then analyzed with the aforementioned standard algorithms (logical regression, k nearest neighbors, decision tree and neural networks). In addition to aforementioned MLP, we used convolutive networks (CNNs), since they are thought to be well suited to image analysis [12,25] and are currently considered the gold standard [61]. In addition to conventional so-called dense layers, CNNs involve convolutional layers where each neuron is stimulated by a restricted set of neurons belonging to the underlying layer through a translation-independent set of weights (*kernels*). Also, a given layer may directly stimulate numerous upper layers (*feature maps*) with different sets of weights (*filters*). This architecture allows the model to identify motives of growing complexity in a fairly hierarchical way. Further, so-called dropout layers appeared to be a powerful means of reducing overfitting. The Tensorflow platform was used, taking advantage of the keras application programming interface. A number of architectures were tested by modification of a number of parameters (number of filters, kernel size, addition of an input channel for simultaneous processing of FITC and DAPI images, activation and loss parameter) starting from a suggested simple architecture ([25] p. 496). However, due to the high number of parameters and long training time, these attempts remained preliminary.Under all conditions, efficiency parameters were calculated by random splitting of datasets between 10 and 100 times into a training set (about 75% of samples) and a testing set (about 25% of samples). Classification efficiency was then calculated on the training and testing set after training models on training sets.

As shown in Figure 8, when 32 different models and model settings were used to calculate all four indices *in the same dataset* of 1733 cells, a tight correlation was found between these indices.

### 4.6. Statistics

The significance of efficiency indices yielded by different models was calculated with Student’s *t*-test in two-tail mode and using Satterthwaite’s correction for degree of freedom determination [62]. Calculations were performed with Libre Office statistical tools (http://www.libreoffice.org (accessed on 20 January 2024)).

## 5. Conclusions

In addition to the description of a simple ML model that performed positive/negative ANCA discrimination matching experienced human analysis after training on a limited dataset, the main conclusion of this report is that simple models such as KNN may be more rewarding than complex neural networks in the performance of simple classification tasks with limited datasets. Indeed, model settings are easier to select with simple models; also, results obtained with conceptually simple models may be more easily interpreted that those yielded by models as complex as neural networks, which may involve more than one million parameters [63], which might facilitate further progress. This conclusion may be an incentive to address problems of medical interest with ML, even when it is not feasible to launch rapidly a large-scale project, involving the building of a large dataset and operation of complex analytical tools.

## Figures and Tables

**Figure 1 ijms-25-03270-f001:**
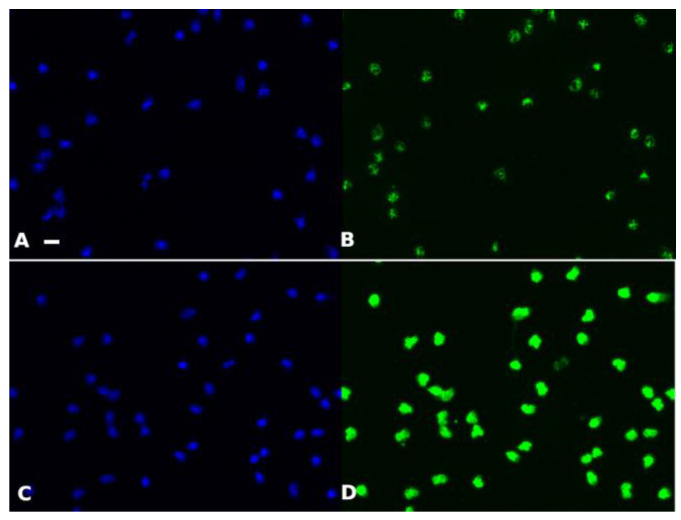
Representative microscopic images. Ethanol-fixed neutrophils were processed for immunofluorescence with a serum positive for C-ANCA (**A**,**B**) or P-ANCA (**C**,**D**) and DAPI (**A**–**C**) or FITC (**C**,**D**) were revealed by fluorescence microscopy. Bar = 10 µm.

**Figure 2 ijms-25-03270-f002:**
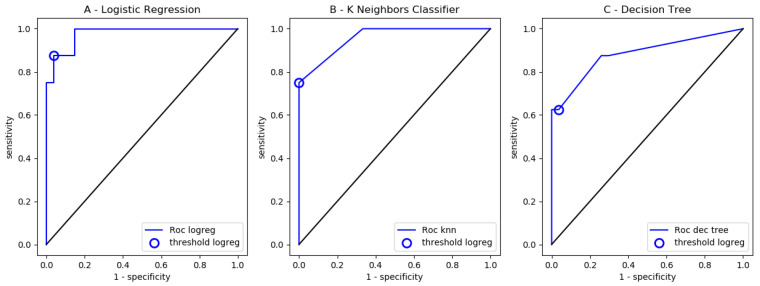
*ROC curves*. With indirect immunofluorescence, 137 sera were assayed for ANCAs and classified as negative (102/137) or positive (35/137) after conventional reading by an experienced biologist. Digitized images of microscopic fields were processed with a computerized algorithm yielding 4 quantitative parameters. The obtained dataset was then randomly split between a training set (102 images) and a testing set (35 images). ROC auc parameters were, respectively, 0.856 (**A**), 0.875 (**B**) and 0.838 (**C**).

**Figure 3 ijms-25-03270-f003:**
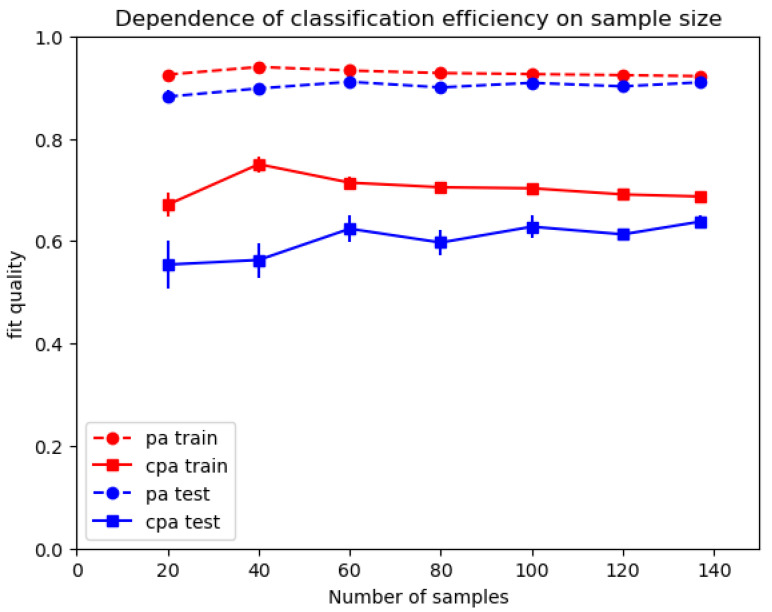
*Dependence of index-based classification efficiency on dataset size.* A dataset of 137 serum samples was randomly reduced to 20, 40, 60, 80, 100 or 120 samples, and the efficiency of positive/negative discrimination by LR was calculated. This process was repeated 100 times for each sample number, and mean *cpa* is shown. Vertical bar length is twice the SEM.

**Figure 4 ijms-25-03270-f004:**
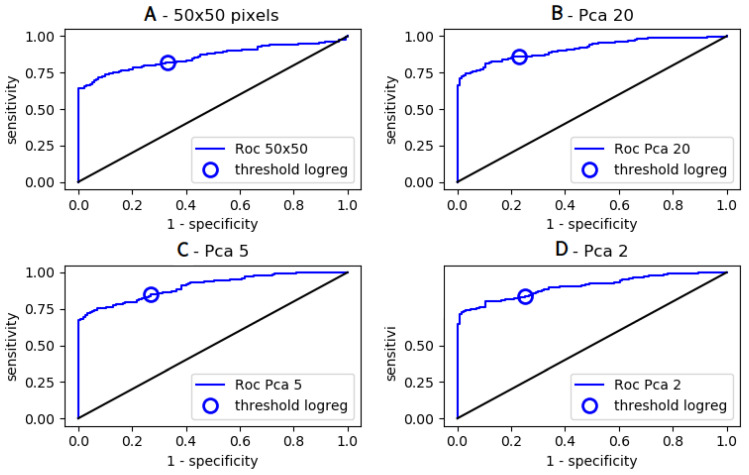
*Pos/Neg discrimination of 2500-pixel images with logistic regression*. A set of 1733 cell images (1200 positive, 533 negative) was randomly split into a training set and a testing set for pos/neg discrimination. Principal component analysis was used for data reduction by retaining 20, 5 or 2 components. ROC curves obtained with LR are shown ((**A**): 2500 parameters, (**B**): 20 components, (**C**): 5 components, (**D**): 2 components).

**Figure 5 ijms-25-03270-f005:**
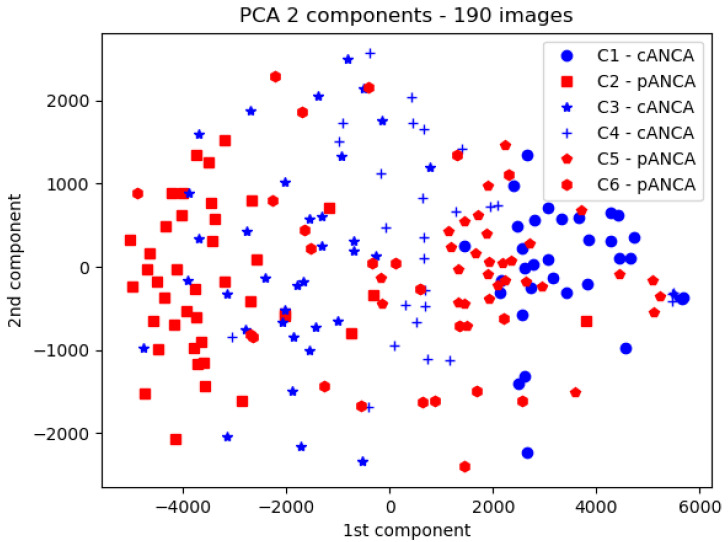
*PCA visualization of the clustering of cell images labelled with the same serum*. Six microscope images generated with 6 sera (3 C-ANCA, 3 P-ANCA) yielded 190 individual cell images. Pixel intensities were subjected to principal component analysis, and the first two components are displayed. Clearly, C-ANCA (blue) and P-ANCA (red) displayed marked overlap, but images corresponding to the same serum displayed significant separation.

**Figure 6 ijms-25-03270-f006:**
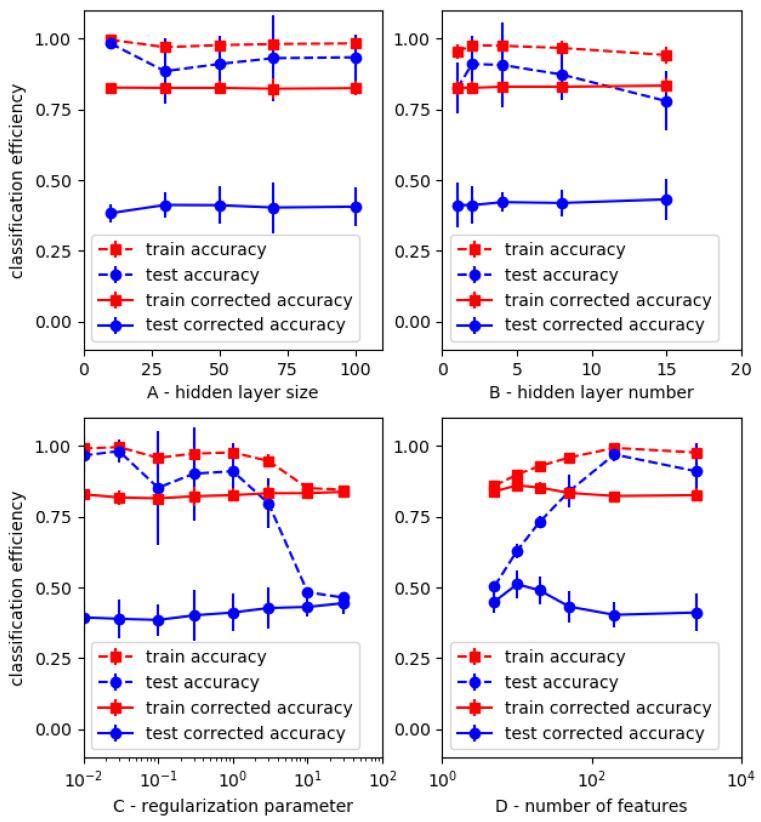
*Dependence of MLP efficiency on model settings.* A series of neural networks with varying number and size of hidden layers and regularization parameters were used to discriminate between positive (533) and negative (1200) samples in a set of 1733 images. For each hyperparameter combination, this set was randomly split between 10 and 40 times into a training and testing set. In another series of calculations, PCA was used for data reduction. Mean values of efficiency parameters are shown. Vertical bar line is twice the standard error.

**Figure 7 ijms-25-03270-f007:**
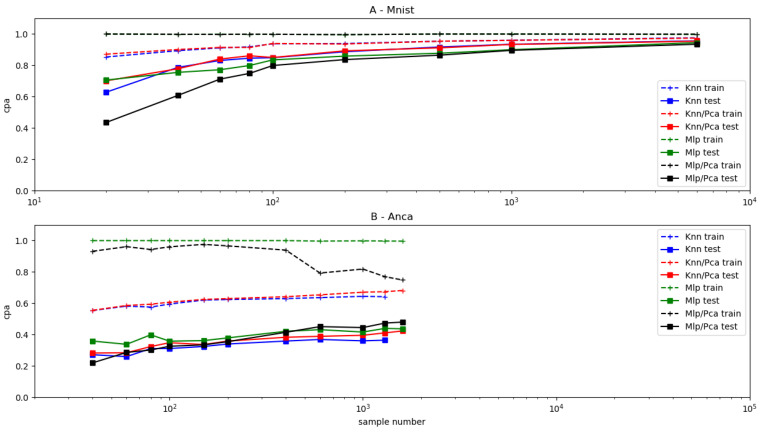
*Dependence of binary classification on sample number*. Two ML models, KNN (3 neighbors) and MLP (two 50-node hidden layers), were used to process two datasets for binary classification. (**A**)—12,000 images (784 pixels each) representing two fashion items (T-shirts or trousers) were extracted from the fashion-MNIST public database. (**B**)—1733 images (2500 pixels) representing cells (positive or negative) immunolabelled for ANCA detection were processed (i) with all pixel intensities or (ii) with twenty major components as obtained with principal component analysis. Both datasets were randomly split one hundred times between a training and a testing set, and mean values of corrected prediction accuracy (*cpa*) obtained on train and test data are shown.

**Figure 8 ijms-25-03270-f008:**
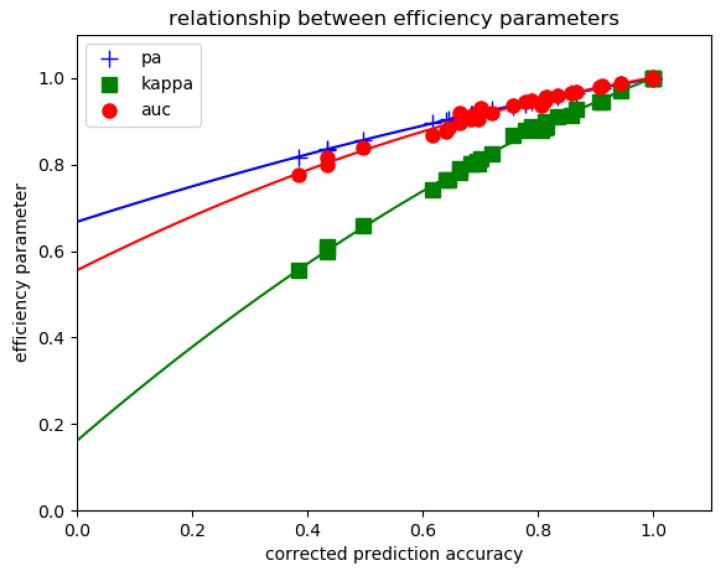
Relationship between classification efficiency indices. A series of 1733 cell images were classified with two models (k neighbors classifier and multilayer perceptron classifier) using different hyperparameters and different preprocessing procedures (data scaling with or without dimensional reduction based on principal component analysis). Prediction accuracy (*pa*), area under ROC curve (*auc*) and Cohen kappa score were plotted versus corrected prediction accuracy (*cpa*) as shown.

**Table 1 ijms-25-03270-t001:** Discrimination between ANCA-positive and -negative sera by processing a 4-parameter dataset.

Analytic Tool	Dataset	Prediction Accuracy (*pa*)	Corrected Prediction Accuracy (*cpa*)	Area under ROC Curve (*auc*)
Logistic regression	train	0.92 +/− 0.002	0.68 +/− 0.006	0.95 +/− 0.001
	test	0.91 +/− 0.004	0.64 +/− 0.014	0.95 +/− 0.004
Nearest neighbors (3 neighbors)	training	0.93 +/− 0.002	0.73 +/− 0.007	0.88 +/− 0.003
	testing	0.89 +/− 0.005	0.56 +/− 0.015	0.81 +/− 0.007
Decision tree (maximum depth: 3)	training	0.96+/− 0.002	0.83 +/− 0.006	0.94 +/− 0.003
	testing	0.89 +/− 0.005	0.56 +/− 0.016	0.84 +/− 0.007

With indirect immunofluorescence, 137 sera were assayed for ANCAs and classified as negative (102/137) or positive (35/137) after conventional reading by an experienced biologist. Digitized images of microscopic fields were processed with a computerized algorithm yielding 4 quantitative parameters. The obtained dataset was then randomly split 100 times between a training set (102 images) and a testing set (35 images). The classification efficiency of three standard classifiers was then assayed on the training and testing sets after training on the training set. Mean results of accuracy indices are shown +/− standard error of the mean.

**Table 2 ijms-25-03270-t002:** Discrimination by ML processing of 4 indices between cytoplasmic and nuclear patterns.

Analytic Tool	Dataset	Prediction Accuracy	Corrected Accuracy	Area under ROC Curve (auc)
Logistic regression	training	0.84 +/− 0.006	0.32 +/− 0.024	0.79 +/− 0.006
	testing	0.77 +/− 0.013	0.17 +/− 0.030	0.78 +/− 0.030
Nearest neighbors (3 neighbors)	training	0.85 +/− 0.004	0.39 +/− 0.016	0.71 +/− 0.008
	testing	0.79 +/− 0.011	0.23 +/− 0.028	0.66 +/− 0.015
Decision tree (maximum depth: 3)	training	0.94 +/− 0.004	0.73 +/− 0.016	0.91 +/− 0.006
	testing	0.68 +/− 0.014	0.04 +/− 0.018	0.57 +/− 0.017

Thirty-five ANCA-positive sera were concluded to yield a cytoplasmic (9/35) or perinuclear/nuclear (26/35) pattern after conventional reading by an experienced biologist. Digitized images of microscopic fields were processed with a computerized algorithm yielding 4 quantitative parameters. The obtained dataset was then randomly split 100 times between a training set (26/35) and a testing set (9/35). The classification efficiency of three standard classifiers was then assayed. Mean results of accuracy indices are shown +/− standard error of the mean.

**Table 3 ijms-25-03270-t003:** Discrimination between four fluorescence patterns by processing a 4-parameter dataset.

Analytic Tool	Dataset	Prediction Accuracy (*pa*)	Corrected Prediction Accuracy (*cpa*)
Logistic regression	training	0.87 +/− 0.002	0.66 +/− 0.005
	testing	0.82 +/− 0.005	0.61 +/− 0.012
Nearest neighbors (3 neighbors)	training	0.90 +/− 0.002	0.75 +/− 0.006
	testing	0.81 +/− 0.006	0.56 +/− 0.014
Decision tree (maximum depth: 3)	training	0.90 +/− 0.002	0.80 +/− 0.005
	testing	0.79 +/− 0.007	0.57 +/− 0.013

A total of 137 sera were assayed for ANCAs with indirect immunofluorescence and categorized as negative (102/137), C-ANCA (9/137), P-ANCA (21/137) or anti-nuclear (5/137) after conventional reading by an experienced biologist. Digitized images of microscopic fields were processed with a computerized algorithm yielding 4 quantitative parameters. The obtained dataset was then randomly split 100 times between a training set (102 images) and a testing set (35 images). The classification efficiency of three standard classifiers was then assayed on the train and testing sets after training on the training set. Mean results of accuracy indices are shown +/− standard error of the mean.

**Table 4 ijms-25-03270-t004:** Discrimination between ANCA-positive and -negative sera by processing 2500-pixel images.

Number of Parameters	Discrimination Parameter	Logistic Regression	Nearest Neighbors (3 Neighbors/Scaling)	Decision Tree (Maximum Depth 3)
2500 (no pca)	cpa training	1.0 +/− 0.0	0.65 +/− 0.019	0.64 +/− 0.030
	cpa testing	0.31 +/− 0.038	0.35 +/− 0.046	0.35 +/− 0.047
	auc testing	0.86 +/− 0.014	0.76 +/− 0.025	0.78 +/− 0.030
20	cpa training	0.48 +/− 0.015	0.68 +/− 0.019	0.58 +/− 0.018
	cpa testing	0.45+/− 0.043	0.42 +/− 0.045	0.38 +/− 0.047
	auc testing	0.92 +/− 0.011	0.80 +/− 0.021	0.80 +/− 0.025
5	cpa training	0.46 +/− 0.015	0.66 +/− 0.018	0.54 +/− 0.024
	cpa testing	0.45 +/− 0.044	0.39 +/− 0.041	0.40 +/− 0.050
	auc testing	0.92 +/− 0.001	0.79 +/− 0.020	0.81 +/− 0.031
2	cpa training	0.43 +/− 0.016	0.60 +/− 0.017	0.49 +/− 0.018
	cpa testing	0.43 +/− 0.043	0.35 +/− 0.039	0.38 +/− 0.0045
	auc testing	0.91 +/− 0.012	0.77 +/− 0.021	0.80 +/− 0.0027

A total of 1733 images from 51 microscope fields were assayed for ANCAs with indirect immunofluorescence and categorized as negative (513/1733) or positive (1220/1733) after conventional reading by an experienced biologist. Digitized images of microscopic fields (50 × 50 pixels) were analyzed as 2500-parameter objects or preprocessed with principal component analysis for retaining the main 20, 5 or 2 components. Datasets were then randomly split 100 times between a training set (1299 images) and a testing set (434 images). Three simple algorithms were then trained and assayed for discriminative efficiency. Mean values of corrected predictive accuracy (*cpa*) for training and testing datasets and area under ROC curve (*auc*) for testing datasets are shown +/− standard deviation.

**Table 5 ijms-25-03270-t005:** Discriminating between cytoplasmic and nuclear patterns by processing 2500-pixel cell images.

ML Algorithm	Dataset	Prediction Accuracy (pa)	Corrected Accuracy (cpa)	Area under ROC Curve (auc)
Logistic regression	training full	1.00 +/− 0.00 SD	1.00 +/− 0.00 SD	1.00 +/− 0.00 SD
	testing full	0.67 +/− 0.024 SD	0.04 +/− 0.022 SD	0.54 +/− 0.019 SD
	training 20c	0.76 +/− 0.008 SD	0.06 +/− 0.018SD	0.53 +/− 0.009 SD
	testing 20c	0.75 +/− 0.020 SD	0.03 +/− 0.021 SD	0.52 +/− 0.011 SD
Nearest neighbors (3 neighbors, scaling)	training full	0.84 +/− 0.006 SD	0.40 +/− 0.017 SD	0.74 +/− 0.011 SD
	testing full	0.71 +/− 0.022 SD	0.08 +/− 0.034 SD	0.55 +/− 0.025 SD
	training 20c	0.85 +/− 0.007 SD	0.42 +/− 0.023 SD	0.74 +/− 0.012 SD
	testing 20c	0.73 +/− 0.019 SD	0.11 +/− 0.033 SD	0.58 +/− 0.022 SD
Decision tree (maximum depth: 5)	training full	0.83 +/− 0.015 SD	0.34 +/− 0.032 SD	0.68 +/− 0.33 SD
	testing full	0.73 +/− 0.023 SD	0.09 +/− 0.038 SD	0.56 +/− 0.023 SD
	training 20c	0.81 +/− 0.013 SD	0.28 +/− 0.048 SD	0.65 +/− 0.036 SD
	testing 20c	0.55 +/− 0.023 SD	0.07 +/− 0.036 SD	0.55 +/− 0.023 SD

A total of 1220 cell images classified as C-ANCA (309/1220) or with a nuclear/perinuclear pattern (911/1220) after conventional reading by an experienced biologist were processed with three standard ML algorithms. The obtained dataset was then randomly split 100 times between a training set (915/1220) and a testing set (305/1220). The classification efficiency of three standard classifiers was then calculated either on full sets of pixel intensities (2500 pixels per image) or using the first 20 components yielded by principal component analysis. Mean results of accuracy indices are shown +/− standard deviation (SD).

**Table 6 ijms-25-03270-t006:** Discrimination between ANCA-positive and -negative sera by processing of individual cell images.

Model	Feature Number	Controlled Cell Splitting	Prediction Accuracy (pa)	Corrected Accuracy (cpa)	Cohen Kappa Score
KNN	2500	No	0.77 +/− 0.01 SE	0.28 +/− 0.03 SE	0.45 +/− 0.02 SE
		Yes	0.77 +/− 0.02 SE	0.28 +/− 0.05 SE	0.44 +/− 0.05 SE
	20	No	0.79 +/− 0.01 SE	0.32 +/− 0.02 SE	0.49 +/− 0.02 SE
		Yes	0.89 +/− 0.02 SE	0.57 +/− 0.05 SE	0.71 +/− 0.04 SE
MLP	2500	No	082 +/− 0.01 SE	0.39 +/− 0.03 SE	0.54 +/− 0.03 SE
		Yes	0.91 +/− 0.02 SE	0.67 +/− 0.06 SE	0.79 +/− 0.04 SE
	20	No	0.82 +/− 0.01 SE	0.39 +/− 0.02 SE	0.58 +/− 0.02 SE
		Yes	0.94 +/− 0.01 SE	0.74 +/− 0.05 SE	0.84 +/− 0.03 SE

Fifty-one (35 positive, 16 negative) sera were tested for ANCA detection with immunofluorescence: 1733 cell images were processed with two machine learning algorithms, k-nearest neighbors (KNN, n = 3 neighbors) and multilayer perceptron (MLP, one 40-neuron hidden layer), for positive/negative discrimination. The image dataset was randomly split either without any restriction (no control) or while ensuring that all images generated with a given sera were gathered in the same (training or testing) group. This process was repeated 25-fold, and mean values of efficiency parameters are displayed together with standard error of the mean (SE). The prediction accuracy was calculated either for each cell (no control group) or for each serum, by classifying each serum as the most frequent classification of corresponding cell images.

## Data Availability

All data used in the present study will be communicated on request after proper agreement of the Institutional Ethical Board.

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
