# Peer review of "Comparison of the Capacity of Several Machine Learning Tools to Assist Immunofluorescence-Based Detection of Anti-Neutrophil Cytoplasmic Antibodies"

_ijms, 2024, doi:10.3390/ijms25063270_

Round 1
Reviewer 1 Report
Comments and Suggestions for Authors
The manuscript "Comparison of the Capacity of Several Machine Learning Tools to Assist Immunofluorescence-Based Detection of Anti-Neutrophil Cytoplasmic Antibodies" investigates the application of machine learning (ML) algorithms to improve the detection of ANCAs, essential for diagnosing autoimmune vasculitides. This study evaluates various ML models, including logistic regression, k-nearest neighbors, decision trees, multilayer perceptrons, and convolutional neural networks, using a dataset of 1733 cell images from 137 sera. Key findings demonstrate that ML models can effectively distinguish ANCA-positive from ANCA-negative samples, achieving substantial agreement with human experts, highlighted by a high kappa score.
However, the study encounters challenges in differentiating between specific fluorescence patterns, underscoring the complexity of automated image analysis and the need for larger datasets and advanced ML strategies. The comparison between simple and complex models reveals that simpler models might be preferable for limited datasets due to their interpretability and reduced overfitting risk.
The manuscript emphasizes the potential of ML to streamline diagnostics by quickly identifying negative samples, thus expediting further testing. It also discusses the necessity for dataset expansion and independent validation to enhance model reliability. Overall, the study provides significant insights into leveraging ML for the clinical diagnosis of autoimmune diseases, indicating a promising direction for future research in integrating AI with clinical diagnostics.
To enhance the overall quality of your manuscript, it is recommended to consider the following enhancements:
Introduction
1. The introduction provides a comprehensive overview of the relevance of AI and ML in enhancing diagnostic precision in medicine, particularly through the analysis of immunofluorescence images for ANA and ANCA detection. However, the transition between paragraphs could be smoother to guide the reader more effectively through the narrative. Consider introducing headings or markers that delineate the subsections, such as the background, the challenge of immunofluorescence, previous efforts in automation, and the potential of ML.
2. While the manuscript references a wide range of sources to support its statements, it could benefit from including more recent studies to highlight the current state of AI and ML in medical diagnostics. This would demonstrate the ongoing advancements in the field and how the current work fits into the broader research context.
3. The manuscript successfully outlines the potential of ML tools in classifying immunofluorescence images but could provide more detail on the specific algorithms and models tested. For example, including brief descriptions of how logistic regression, K neighbor classifiers, and decision trees were applied and why these particular models were chosen could offer valuable insights to the reader. Additionally, mentioning any challenges or limitations encountered with these models would enhance the discussion on the need for more complex neural networks.
4. The section briefly mentions the dataset used (1733 cell images obtained from 137 sera) but lacks detail about the dataset's composition and how it was curated. Expanding on how the images were selected, any preprocessing steps, and the criteria for including sera in the study would strengthen the methodological foundation of the work.
5. While the section is primarily introductory, offering a preview of the results (e.g., mentioning the kappa scores higher than 0.8) is engaging. However, it would be beneficial to more explicitly link these outcomes to the broader goals of the study and the implications for ANCA detection. A clearer statement on how these results compare to existing methods could set the stage for the detailed findings that follow.
6. The mention of "promise and pitfalls" at the end is intriguing but somewhat vague. Providing a brief overview of the specific challenges faced—such as the need for larger datasets or the complexity of pattern classification—and potential strategies for overcoming these issues would offer a more compelling conclusion to the introduction and better transition to the rest of the manuscript.
7. The manuscript is technically dense, which is appropriate for its audience, but ensuring that complex concepts are accessible to readers less familiar with ML could broaden its impact. Consider adding brief explanations or definitions for some of the more technical terms and acronyms upon first use.
Results
8. The detailed presentation of results, including predictive accuracy, corrected prediction accuracy, and area under the ROC curve for different ML algorithms, provides a solid foundation for evaluating the efficacy of these approaches. It is commendable how the study systematically explores different algorithms and settings, including logistic regression, k-nearest neighbors, and decision trees, across various dataset configurations.
9. The statistical analysis, including standard errors and deviations, adds rigor to the findings. However, it would be beneficial to discuss the statistical significance of differences observed between algorithms and between training and test sets more explicitly. This could involve statistical tests comparing the performance of different models or discussing the confidence intervals of performance metrics to better understand their reliability and the potential for overfitting.
10. While the manuscript provides a good overview of the ML algorithms used, additional methodological details could enhance reproducibility and understanding. For instance, elaborating on the feature selection process, the rationale behind choosing specific hyperparameters, and any preprocessing steps applied to the data would be valuable.
11. The manuscript could benefit from a more direct comparison with existing diagnostic methods or ML approaches. Discussing how the proposed ML models compare to traditional diagnostic criteria or to other automated systems in terms of efficiency, cost, and scalability would provide valuable context for the significance of the findings.
12. The discussion around the limitations of the study, particularly regarding dataset size and the discriminative power of the chosen features, is important. Expanding on these limitations and outlining specific future research directions that could address them would strengthen the manuscript. For example, discussing the potential for integrating more diverse datasets, exploring additional features, or employing more complex neural network architectures could offer insights into how to overcome current challenges.
13. The manuscript would benefit from a clearer articulation of the clinical implications of these findings. Discussing how these ML approaches could be integrated into clinical practice, the potential impact on patient outcomes, and any ethical considerations associated with automated diagnosis could make the research more relevant to a broader audience.
14. The inclusion of figures, such as the principal component analysis visualization, is effective in illustrating key points. Additional visual aids, such as ROC curves for each ML model or heatmaps of feature importance, could further enhance the reader's understanding of the model's performance and decision-making process.
15. The manuscript does a commendable job of investigating the impact of neural network hyperparameters on model performance. Highlighting the moderate dependence of classification efficiency on hyperparameter variation is insightful. However, it would be beneficial to provide more detailed guidance on how the optimal hyperparameter settings were determined, potentially including a discussion on the use of techniques like grid search or random search for hyperparameter optimization.
16. The comparison of neural network performance against standard machine learning models (as shown in Table 4) is valuable. It clearly demonstrates the superior capability of neural networks in handling complex image data. Elaborating on why neural networks outperform these models, possibly by discussing their ability to capture hierarchical features in the images, could enrich the analysis.
17. The manuscript identifies overfitting as a significant challenge, especially when dealing with high-dimensional data without sufficient regularization. It would be useful to discuss strategies that were employed or could be employed to mitigate overfitting, such as dropout, data augmentation, or early stopping, and their potential impact on model performance.
18. The innovative approach of controlled splitting to ensure all images from a given serum are in the same dataset partition, and classifying sera based on the majority vote of cell image classifications, represents a novel strategy. Discussing the implications of this approach for clinical practice, where serum-based diagnosis is standard, could highlight its practical significance. Further exploration of how this strategy might be optimized or combined with other techniques for even greater accuracy would also be informative.
19. The limited success in pattern classification using MLPs and the lack of improvement with CNNs suggest a fundamental challenge in extracting and utilizing pattern-specific features from the images. A deeper dive into this issue, possibly including a discussion on the limitations of the feature sets used and exploring more advanced neural network architectures or feature extraction techniques, could be enlightening.
20. Given the challenges encountered in improving pattern classification and the modest improvements achieved through controlled dataset splitting, outlining specific future research directions could be beneficial. This might include exploring more complex neural network architectures like deeper CNNs, integrating additional types of data (e.g., patient clinical data), or employing transfer learning and fine-tuning with pre-trained networks.
Discussion
21. You highlight the potential clinical benefits of ML in reducing diagnostic delays through efficient positive/negative discrimination of ANCA-associated conditions. Expanding on how this technological advancement could be integrated into current diagnostic workflows and its potential impact on patient management and outcomes would offer valuable insights into the practical application of your findings.
22. The manuscript rightly points out the importance of validation using an independent dataset to ensure the reliability of the ML models. It might be beneficial to further discuss potential strategies for obtaining and utilizing such datasets, including collaborations with other laboratories or institutions, to enhance the generalizability of the models.
23. Your acknowledgment of the challenges in discriminating between different fluorescence patterns sets the stage for future research. Discussing in more detail the technical or methodological barriers that currently prevent sophisticated pattern recognition and how emerging technologies or methodologies might overcome these limitations could be enlightening.
24. The observation that simpler models like kNN may outperform more complex neural networks in certain scenarios is intriguing. Delving deeper into the reasons behind this—such as the interpretability of simpler models, the challenges of overfitting with complex models, and the trade-offs between model complexity and dataset size—would provide a more nuanced understanding of when and why to choose one approach over another.
25. The use of biologically inspired indices as a bridge between biological expertise and AI is a significant insight. Discussing potential new indices that could be developed from a deeper biological understanding of the diseases in question, or how existing indices might be optimized for ML purposes, could highlight paths for innovation.
26. The call for database expansion is well justified. Providing examples of how specific classification errors were analyzed to refine the model or how additional data types (e.g., clinical data, patient outcomes) could be incorporated to improve model performance would be beneficial. Moreover, discussing the logistical and ethical considerations in expanding databases, such as data privacy and patient consent, could add depth to the discussion.
27. Finally, outlining more specific future research directions based on the findings and limitations of the current study would be helpful. This could include the exploration of hybrid models that combine the strengths of both simple and complex ML approaches, the development of novel data augmentation techniques to enrich limited datasets, or the application of transfer learning from related domains.
Comments on the Quality of English LanguageThe English quality, based on the provided excerpts and summaries, appears to be generally good, with complex ideas and research findings communicated effectively. However, without reviewing the full text, specific detailed feedback on grammar, syntax, or stylistic nuances is limited.
Author Response
Answer to reviewers
We thank the reviewers for their careful reading of the manuscript and interesting remarks. We agree
that we essentially focussed on the description of fairly simple and older methods, since our aim was
only to show that these simple methods could yield significant results.
We agree that it was useful to give more information on recent advances and provide a more precise
description of available strategies to improve classification. The manuscript was thus thoroughly
corrected. In order to keep it readable to a medical audience, we located a substantial part of
information first in the method section and also in the discussion, since we felt that some details would
be more interesting for a reader having encountered specific problems.
To facilitate the reviewers's task, we added line numbers to the PDF version of the corrected file, and
significant changes were printed in red.
Reviewer #1
Comments and Suggestions for Authors
The manuscript "Comparison of the Capacity of Several Machine Learning Tools to Assist
Immunofluorescence-Based Detection of Anti-Neutrophil Cytoplasmic Antibodies" investigates the
application of machine learning (ML) algorithms to improve the detection of ANCAs, essential for
diagnosing autoimmune vasculitides. This study evaluates various ML models, including logistic
regression, k-nearest neighbors, decision trees, multilayer perceptrons, and convolutional neural
networks, using a dataset of 1733 cell images from 137 sera.
Key findings demonstrate that ML models can effectively distinguish ANCA-positive from ANCA-
negative samples, achieving substantial agreement with human experts, highlighted by a high kappa
score. However, the study encounters challenges in differentiating between specific fluorescence
patterns, underscoring the complexity of automated image analysis and the need for larger datasets
and advanced ML strategies. The comparison between simple and complex models reveals that simpler
models might be preferable for limited datasets due to their interpretability and reduced overfitting
risk.
The manuscript emphasizes the potential of ML to streamline diagnostics by quickly identifying
negative samples, thus expediting further testing. It also discusses the necessity for dataset expansion
and independent validation to enhance model reliability.
Overall, the study provides significant insights into leveraging ML for the clinical diagnosis of
autoimmune diseases, indicating a promising direction for future research in integrating AI with
clinical diagnostics.
To enhance the overall quality of your manuscript, it is recommended to consider the following
enhancements:
Introduction
1. The introduction provides a comprehensive overview of the relevance of AI and ML in enhancing
diagnostic precision in medicine, particularly through the analysis of immunofluorescence images for
ANA and ANCA detection. However, the transition between paragraphs could be smoother to guide the
reader more effectively through the narrative. Consider introducing headings or markers that delineate2/9
the subsections, such as the background, the challenge of immunofluorescence, previous efforts in
automation, and the potential of ML.
We thank the reviewer for this suggestion that we gladly followed. Also, the introduction was
thoroughly corrected to increase readability and add useful quantitative information.
2. While the manuscript references a wide range of sources to support its statements, it could benefit
from including more recent studies to highlight the current state of AI and ML in medical diagnostics.
This would demonstrate the ongoing advancements in the field and how the current work fits into the
broader research context.
We added more references in the introduction, and also in the discussion to illustrate possible strategies
to increase the scope and validity of presented models. Indeed, the total number of references was
increased by more than 50%.
3. The manuscript successfully outlines the potential of ML tools in classifying immunofluorescence
images but could provide more detail on the specific algorithms and models tested. For example,
including brief descriptions of how logistic regression, K neighbor classifiers, and decision trees were
applied and why these particular models were chosen could offer valuable insights to the reader.
Additionally, mentioning any challenges or limitations encountered with these models would enhance
the discussion on the need for more complex neural networks.
We tried to provide a simple description of the ML models we used in the Methods section (see lines
256-270). The choise followed the conclusions of a former study performed with a limited dataset
(reference 34 of revised paper)
4. The section briefly mentions the dataset used (1733 cell images obtained from 137 sera) but lacks
detail about the dataset's composition and how it was curated. Expanding on how the images were
selected, any preprocessing steps, and the criteria for including sera in the study would strengthen the
methodological foundation of the work.
This information was added in section 2.1 (line 160) and section 2.2 (lines 169-170). Details on
preprocessing were added in section 2.4.2 lines 256-261).
5. While the section is primarily introductory, offering a preview of the results (e.g., mentioning the
kappa scores higher than 0.8) is engaging. However, it would be beneficial to more explicitly link these
outcomes to the broader goals of the study and the implications for ANCA detection. A clearer
statement on how these results compare to existing methods could set the stage for the detailed findings
that follow.
In the introduction, we indicated currently obtained kappa scores for ANCAs (line 128) et for ANAs
that were more often studied (134). As pointed out in the discussion, a kappa score higher than 0.8 is
considered as "very good" (line 698).
6. The mention of "promise and pitfalls" at the end is intriguing but somewhat vague. Providing a brief3/9
overview of the specific challenges faced—such as the need for larger datasets or the complexity of
pattern classification—and potential strategies for overcoming these issues would offer a more
compelling conclusion to the introduction and better transition to the rest of the manuscript.
We agree that this mention was too vague. The introduction was corrected. We briefly indicated that
potential strategies would be desribed in the discussion, since we felt that the introduction was long
enough, and the detailed description of the strategy would better fit into the discussion, that should be
clearer after reading the methods and results section.
7. The manuscript is technically dense, which is appropriate for its audience, but ensuring that
complex concepts are accessible to readers less familiar with ML could broaden its impact. Consider
adding brief explanations or definitions for some of the more technical terms and acronyms upon first
use.
We tried to provide simple definitions for important terms in the methods section (lines 256-273)
Results
8. The detailed presentation of results, including predictive accuracy, corrected prediction accuracy,
and area under the ROC curve for different ML algorithms, provides a solid foundation for evaluating
the efficacy of these approaches. It is commendable how the study systematically explores different
algorithms and settings, including logistic regression, k-nearest neighbors, and decision trees, across
various dataset configurations.
We thank the reviewer for this remark.
9. The statistical analysis, including standard errors and deviations, adds rigor to the findings.
However, it would be beneficial to discuss the statistical significance of differences observed between
algorithms and between training and test sets more explicitly. This could involve statistical tests
comparing the performance of different models or discussing the confidence intervals of performance
metrics to better understand their reliability and the potential for overfitting.
We added a number of P values (calculated as indicated in the new section 2.5) to give a quantitative
account of the statistical significance of results displayed on Tables.
10. While the manuscript provides a good overview of the ML algorithms used, additional
methodological details could enhance reproducibility and understanding. For instance, elaborating on
the feature selection process, the rationale behind choosing specific hyperparameters, and any
preprocessing steps applied to the data would be valuable.
We tried to give more detailed information on models in the method section. The data set structure is
fully described. Models were chosen on the basis of a previous paper devoted to the analysis of a
limited dataset (reference 34 of the new manuscript), and we used scikit-learn default values of
hyperparameters (line 256). The low values of max_depth and n_neighbors were suggested by ref 34.
However, we checked that this choice did not result in important efficacy changes (not shown and
Figure 7).4/9
11. The manuscript could benefit from a more direct comparison with existing diagnostic methods or
ML approaches. Discussing how the proposed ML models compare to traditional diagnostic criteria or
to other automated systems in terms of efficiency, cost, and scalability would provide valuable context
for the significance of the findings.
We inserted a maximum number of reported efficiency parameters in the paper, but it must be pointed
out that (i) different authors may use different parameters, (ii) obtained values are dependent on
patient's population and reference data, which sets a limit on the comparison of these parameters.
Systematic comparisons of commercial models were quoted (especially ref 37). However, although the
precise algorithms used by commercial models are not always known, our feeling was that the
"traditional" strategy of a two-step procedure (hand-crafting texture parameters et classification) tended
to be outperformed by deep learning commercial systems (see references 18 and 20 and line 756).
12. The discussion around the limitations of the study, particularly regarding dataset size and the
discriminative power of the chosen features, is important. Expanding on these limitations and outlining
specific future research directions that could address them would strengthen the manuscript. For
example, discussing the potential for integrating more diverse datasets, exploring additional features,
or employing more complex neural network architectures could offer insights into how to overcome
current challenges.
The discussion of available strategies was enlarged and located in the discussion. The interest of dataset
integration was mentioned, but it was indicated that this might reduce clinical benefit (lines 735-739).
The power of recent neural network architectures is emphasized (lines 762-768, see also line 798).
13. The manuscript would benefit from a clearer articulation of the clinical implications of these
findings. Discussing how these ML approaches could be integrated into clinical practice, the potential
impact on patient outcomes, and any ethical considerations associated with automated diagnosis could
make the research more relevant to a broader audience.
The main clinical interest of automated ANCA diagnosis (i.e. delay and cost reduction) is explained on
lines 696-702. The expected benefit in other domains was briefly outlined (lines 801-806).
14. The inclusion of figures, such as the principal component analysis visualization, is effective in
illustrating key points. Additional visual aids, such as ROC curves for each ML model or heatmaps of
feature importance, could further enhance the reader's understanding of the model's performance and
decision-making process.
We showed ROC curves on the new figures 3 & 5
15. The manuscript does a commendable job of investigating the impact of neural network
hyperparameters on model performance. Highlighting the moderate dependence of classification
efficiency on hyperparameter variation is insightful. However, it would be beneficial to provide more
detailed guidance on how the optimal hyperparameter settings were determined, potentially including
a discussion on the use of techniques like grid search or random search for hyperparameter
optimization.5/9
Hyperparameter optimization is mentioned in the new discussion (lines 771-780).
16. The comparison of neural network performance against standard machine learning models (as
shown in Table 4) is valuable. It clearly demonstrates the superior capability of neural networks in
handling complex image data. Elaborating on why neural networks outperform these models, possibly
by discussing their ability to capture hierarchical features in the images, could enrich the analysis.
This point is discussed in lines 292 to 302.
17. The manuscript identifies overfitting as a significant challenge, especially when dealing with high-
dimensional data without sufficient regularization. It would be useful to discuss strategies that were
employed or could be employed to mitigate overfitting, such as dropout, data augmentation, or early
stopping, and their potential impact on model performance.
The effect of regularization is displayed on Figure 7C and lines 428-430, 606-607, 779-781,
data augmentation is discussed on lines 740-744 and dropout on line 297. We tried to make the order as
logical as possible, but it was felt difficult to discuss in a single paper the multiple properties of
complex algorithms.
18. The innovative approach of controlled splitting to ensure all images from a given serum are in the
same dataset partition, and classifying sera based on the majority vote of cell image classifications,
represents a novel strategy. Discussing the implications of this approach for clinical practice, where
serum-based diagnosis is standard, could highlight its practical significance. Further exploration of
how this strategy might be optimized or combined with other techniques for even greater accuracy
would also be informative.
Currently, we think that the interest of this procedure is only to improve diagnosis efficiency. We need
more information on the heterogeneity of (commercial) neutrophil slides to determine whether this
might be used to influence diagnosis and clinical practice. In a second step, it might be informative to
analyse results together with clinical data, but this would require to set a clinical trial.
19. The limited success in pattern classification using MLPs and the lack of improvement with CNNs
suggest a fundamental challenge in extracting and utilizing pattern-specific features from the images. A
deeper dive into this issue, possibly including a discussion on the limitations of the feature sets used
and exploring more advanced neural network architectures or feature extraction techniques, could be
enlightening.
The new Figure 8 was meant to strengthen the conclusion that there is a need to extend the dataset.
Also, we think that the lack of improvement with CNNs is not definitive. Admittedly (as emphasized
on line 302), this part of the study was quite preliminary and did not fit into the main purpose of the
paper.
20. Given the challenges encountered in improving pattern classification and the modest improvements
achieved through controlled dataset splitting, outlining specific future research directions could be
beneficial. This might include exploring more complex neural network architectures like deeper CNNs,6/9
integrating additional types of data (e.g., patient clinical data), or employing transfer learning and
fine-tuning with pre-trained networks.
We tried to present possible strategies in the revised discussion (lines 719-800). We agree that deeper
CNNs would give better results, but we shall first increase the dataset.
Discussion
21. You highlight the potential clinical benefits of ML in reducing diagnostic delays through efficient
positive/negative discrimination of ANCA-associated conditions. Expanding on how this technological
advancement could be integrated into current diagnostic workflows and its potential impact on patient
management and outcomes would offer valuable insights into the practical application of your
findings.
This is discussed in the corrected discussion (lines 696-702)
22. The manuscript rightly points out the importance of validation using an independent dataset to
ensure the reliability of the ML models. It might be beneficial to further discuss potential strategies for
obtaining and utilizing such datasets, including collaborations with other laboratories or institutions,
to enhance the generalizability of the models.
The need for further validation and possible strategies is discussed on lines 711-714
23. Your acknowledgment of the challenges in discriminating between different fluorescence patterns
sets the stage for future research. Discussing in more detail the technical or methodological barriers
that currently prevent sophisticated pattern recognition and how emerging technologies or
methodologies might overcome these limitations could be enlightening.
Possible strategies were discussed in an important part of the discussion (lines 719-800) as explained
above. It must be pointed out (line 801-806) that an increased accuracy of pattern recognition will need
a more precise definition of "true" aspects.
24. The observation that simpler models like kNN may outperform more complex neural networks in
certain scenarios is intriguing. Delving deeper into the reasons behind this—such as the
interpretability of simpler models, the challenges of overfitting with complex models, and the trade-offs
between model complexity and dataset size—would provide a more nuanced understanding of when
and why to choose one approach over another.
Our feeling is that KNN did not outperform MLP (see Table 6 or Table 1 with cpa=0.56 for KNN et
cpa=0.59 for MLP as indicated in text) and we feel that MLP would easily outperform KNN with a
larger datase. The problem of interpretability is important and was mentionet on line 758.
25. The use of biologically inspired indices as a bridge between biological expertise and AI is a
significant insight. Discussing potential new indices that could be developed from a deeper biological
understanding of the diseases in question, or how existing indices might be optimized for ML purposes,
could highlight paths for innovation.7/9
Our feeling is that progress will be more rapid with deep learning rather than trying to find additional
indices (we tried a few new indices, but this did not work to date) - Indeed, our bet would be that
biological intuition might be more efficiently improved by interpreting successful ML algorithms (line
757-758) than classification might be improved by increasing features.
26. The call for database expansion is well justified. Providing examples of how specific classification
errors were analyzed to refine the model or how additional data types (e.g., clinical data, patient
outcomes) could be incorporated to improve model performance would be beneficial. Moreover,
discussing the logistical and ethical considerations in expanding databases, such as data privacy and
patient consent, could add depth to the discussion.
As indicated in the discussion, while the association of different datatypes is a recognized means of
improving classification efficiency , combining staining procedures would pose no ethical problems
(lines 723-729). Adding clinical data is probably not needed, and this might delay data collection and
hamper the increase of testing rapidity that is the main benefit expected from ML for ANCA testing.
27. Finally, outlining more specific future research directions based on the findings and limitations of
the current study would be helpful. This could include the exploration of hybrid models that combine
the strengths of both simple and complex ML approaches, the development of novel data augmentation
techniques to enrich imited datasets, or the application of transfer learning from related domains.
In the case of ANCA classification, there is probably no need for novel data augmentation techniques,
and random rotation is probably the simplest technique. However, our plan is first to increase data sets
by adding more images from a collection that is already available in our laboratory, since we began
storing images ten years ago, when the ICARE project was started.

Reviewer 2 Report
Comments and Suggestions for Authors
Review on “Comparison of the capacity of several machine learning tools to assist immunofluorescence-based detection of anti-neutrophil cytoplasmic antibodies” for manuscript ID ijms-2873977
In this manuscript the authors describe several machine learning approaches to support medical diagnosis by cell images’ analysis. In the brief introduction authors describe the indirect immunofluorescence tool to identify anti-nuclear antibodies, and various algorithms to neglect human factor using computer vision and machine learning.
As line numbers are missing in source PDF, I cannot specify the particular places of interest, mentioning the section only.
Intro section comments:
[6][6] – duplicated reference
Starting with “It was thus an attractive prospect…” the rest of paragraph is hardly related to the topic, no need to describe the common ML practices. Measuring efficiency should be considered in context of a particular method of classification.
Intro section should cover the current ML tools to analyze cell images to detect anti-nuclear antibodies. Authors mention some relatively old tools from [30] (2014), [25] (2013), [24] (2012) and [29] (2015). Are there any progress on the topic since 2015?
My questions about Results and Discussion:
Discussion section lacking comparison with similar tools. Conclusion such as “our simple approach did not succeed in discriminating between different fluorescence patterns”, shows that result is incomplete and further study is needed.
The last paragraph “A general conclusion of our study…” might be omitted while it declares almost common sense logic.
Methods section comments:
How the “dataset including 1733 cell images” was collected? The raw data is not available, the source of cell culture is unknown.
The structure of used machine learning models is not clear. Why the source images weren’t used? Proper CNNs could classify them with high precision.
The training and test dataset is very small (137 samples). Figure 3 shows no trend line because of limited training set. For such small set the transfer learning approach could help. See further https://doi.org/10.1109/ISBI.2016.7493483
The trained model or source files are not publicly available, so readers cannot re-use them or reproduce the results.
Section 2.4.1: the purpose of using ICARE [25] is not clear.
What are the features for training the models? CSV file is missing.
Some minor corrections to the text (style and spelling):
· “larger databas” → “larger databases”
· “2500 pixel images” means 50x50 pixels?
· “083 +/- 0.006” dot missing
· “java” →”Java”
Author Response
We thank the reviewer for his careful reading of the manuscript and constructive comments
Reviewer #2
Comments and Suggestions for Authors
Review on “Comparison of the capacity of several machine learning tools to assist
immunofluorescence-based detection of anti-neutrophil cytoplasmic antibodies” for manuscript ID
ijms-2873977
In this manuscript the authors describe several machine learning approaches to support medical
diagnosis by cell images’ analysis. In the brief introduction authors describe the indirect
immunofluorescence tool to identify anti-nuclear antibodies, and various algorithms to neglect human
factor using computer vision and machine learning.
As line numbers are missing in source PDF, I cannot specify the particular places of interest,
mentioning the section only.
We agree that it would have been more convenient to add line numbers. This was done in the revised
manuscript to facilitate the reviewers' task.
Intro section comments:
[6][6] – duplicated reference
The error was corrected8/9
Starting with “It was thus an attractive prospect...” the rest of paragraph is hardly related to the topic,
no need to describe the common ML practices. Measuring efficiency should be considered in context of
a particular method of classification.
The introduction was reorganized and we added subtitles, as suggested by Reviewer #1, to make the
flow of arguments more apparent.
Intro section should cover the current ML tools to analyze cell images to detect anti-nuclear
antibodies. Authors mention some relatively old tools from [30] (2014), [25] (2013), [24] (2012) and
[29] (2015). Are there any progress on the topic since 2015?
The main purpose of the first version of the manuscript was to show that significant results could be
obtained with fairly simple and old methods. We agree that it was a good idea to add more recen
references in order to discuss the potential of more recent - and more complex - tools. Accordingly, we
added numerous recent publications.
My questions about Results and Discussion:
Discussion section lacking comparison with similar tools. Conclusion such as “our simple approach
did not succeed in discriminating between different fluorescence patterns”, shows that result is
incomplete and further study is needed.
The discussion section was thoroughly corrected. After discussing the interest of obtained results, we
described available strategies to address pending problem. However, building a larger dataset is a
prerequisite to further progress, and we plan to start this rapidly.
The last paragraph “A general conclusion of our study...” might be omitted while it declares almost
common sense logic.
The last paragraph was removed, and replaced with a (hopefully) more significant conclusion.
Methods section comments:
How the “dataset including 1733 cell images” was collected? The raw data is not available, the source
of cell culture is unknown.
We gave more precise information on serum collection (line 160) and image preparation from
microscope images (lines 241-251).
The structure of used machine learning models is not clear. Why the source images weren’t used?
Proper CNNs could classify them with high precision.
We only resized images to study them with models requiring a fixed number of features. The main
purpose was to study simple models, not CNNs, with a limited dataset. As indicated in the conclusion,
the use of a "light" project may facilitate the exploration of more numerous questions of medical
interest. Positive results would thus pave the way to the elaboration of large scale projects.9/9
The training and test dataset is very small (137 samples). Figure 3 shows no trend line because of
limited training set. For such small set the transfer learning approach could help. See further
ttps://doi.org/10.1109/ISBI.2016.7493483.
We agree that Figure 3 was not satisfactory and the new Figure 8 was added to improve this point.
We discussed transfer learning (line 788) and quoted the paper indicated by the reviewer together with
a more recent paper. Note that the paper by Phan et al. (2016) was quoted in reference 11 of the first
version of our paper.
The trained model or source files are not publicly available, so readers cannot re-use them or
reproduce the results.
We tried to give all needed information on our model. We plan to build a public dataset and make
image files available, but this requires a clear agreement of our institue ethical committee. We are
currently writing a project along this line.
Section 2.4.1: the purpose of using ICARE [25] is not clear.
We tried to clarifiy this point (lines 217-218). ICARE is systematically used in our laboratory to
process immunofluorescence studies. Since it was tentatively applied to ANCA images, it results in the
progressive accumulation of images that could be used in the present study.
What are the features for training the models? CSV file is missing.
The indices calculated by ICARE are described on lines 217-239. And the processing of 50x50 pixel
images is described with more details in the revised manuscript (lines 242-251).
Some minor corrections to the text (style and spelling):
· “larger databas” → “larger databases”
· “083 +/- 0.006” dot missing
· “java” →”Java”
We thank the reviewer for pointing out the typos. They were corrected in the revised manuscript
· “2500 pixel images” means 50x50 pixels?
Yes - this was indicated on line 218.

Round 2
Reviewer 1 Report
Comments and Suggestions for Authors
I want to express my sincere gratitude to the authors for their comprehensive and careful revisions. The manuscript has shown considerable improvement from its initial submission, reflecting the authors' genuine commitment to resolving the previously pointed out concerns.
Comments on the Quality of English LanguageMinor editing of English language required
Author Response
Answer to reviewers' comments on revised paper. “Comparison of the capacity of several machine learning tools to assist immunofluorescence-based detection of anti-neutrophil cytoplasmic antibodies” Manuscript ID :ijms-2873977 Resubmitted manuscript is provided as : - a pdf file with changes printed in red. The numbering of this file will be used in the answer to reviewers. - a docx file with a different ordering of sections, resulting in a change of citation numbering, as requested by the editor on march 1st. (reviewers' comments are printed in italic) Answer to Reviewer #1: I want to express my sincere gratitude to the authors for their comprehensive and careful revisions. The manuscript has shown considerable improvement from its initial submission, reflecting the authors' genuine commitment to resolving the previously pointed out concerns. Comments on the Quality of English Language Minor editing of English language required We thank the reviewer for his kind comment. The manuscript was read several times to remove minor errors and to fit the journal template. Answer to Reviewer #2: I would to thank authors for the improving figures and the manuscript, but some concerns remain unclear. Whole manuscript should be carefully checked because multiple minor errors: different fonts, missing commas and stops, spaces etc. Tables should be formatted according to the journal layout. We thank the reviewer for this comment. The manuscript was repeatedly checked. Indeed, in order to comply as much as possible with the amount of time allotted for revision, we performed only a minimal - admittedly insufficient - check of typing errors and grammatical correctness in the revised file submitted on februray 22. My questions about Results and Discussion: Unfortunately, no raw data (CSV, tabular or source images) are publicly available. I see no problem to submit training dataset and source code along with the article to reproduce the results. We first felt as the reviewer. We also thought this should pose no problem. Unfortunately, ethical rules became more and more severe throughout the world and particularly in France. Therefore, we asked the president of local ethics committe if we were allowed to make public fully anonymous csv files. He replied that we had to submit a formal application. Since the preparation and examination by the committee of this (admittedly quite simple) question will require at least many months, we could not include the file. However, we described our calculation methods as fully as possible, and we will be happy to provide any requested information on coding details. This study lacks novelty, simple models are used and many improvements are suggested in Discussion. This study consisted of addressing an unsolved problem: immunofluorescence image classification for ANCA diagnosis with ML, with an accuracy comparable to that provided by human readers. Novelty consisted of building a new dataset and processing it with standard ML tools. Many experimental works are performed in this manner. Indeed, many biological papers described the study of a new molecule with standard biochemical tools. Admittedly, we did not devise any novel ML method. Note that if this were the case, IJMS might not be the best suited journal to report such a theoretical advance. “common opinion that the success of ML is largely due to the mining of large amounts of data [49]” – that opinion paper has nothing to do with the discriminating between different fluorescence patterns. We think that much evidence supports the view that the effectiveness of ML is highly dependent on the content of input data as well as algorithms in many domains, including discrimination between immunofluorescence patterns. As written in reference [25] "the idea that data matters more than algorithms for complex problems was further popularized by Petr Norvig et al." (this is ref [49]). Also, this idea was suggested in a study of fluorescence classification (ref. [19] quoted in line 79). Thus, we recalled ref 19 and we added ref. 49 on line 722. Most of Discussion section is devoted to future steps of the study: dataset improvement, training process improvement. cell image augmentation etc. Indeed, we think that the discussion section of a research paper addressing a specific problem should include an analysis of the significance of obtained results and suggestions about the "road forward". The conclusion of our study is that we must first increase the dataset, then apply a number of tools, as described. [58] PNAS paper does not related to the cell images processing. This PNAS paper presented a general discussion of ML training strategy that is fully relevant to image processing : the last sentence of the Introduction reads :"in this article we present extensive measurements across image classification datasets and architectures, exposing a common empirical pattern" and the authors certainly felt that the theoretical concepts described in their study should apply to image processing. See also line 798. “Fashion-MNIST database” is questionable choice, it differs completely from the cell images used in the present study. The aim of the figure was to show that the "data hungriness" or ML is connected at the same time to qualitative and quantitative properties of samples. Therefore, we compared two fairly different sets of images of comparable size to provide a visual support to this (not very original) idea. The overall conclusion about the manuscript – looks like a preliminary results with reasonable future stages. ML field is developing very fast, I suggest to review best practices of processing microscope images to compare with the results of this study. Recent practices of processing microscope images with ML are described in depth in reference [12] and recent examples of ANA and ANCA classification were described in the introduction. The aim of our paper was rather to show that useful results could be obtained with limited datasets and fairly simple methods, which should be an incentive for the medical community to try and apply these tools to all questions they feel of interest. Indeed, many (outstanding) successes or AI are obtained with huge dat sets and highly complex models (see line 777). This might deter scientists from studying some problems they feel of interest. This was the message we wished to include. Methods section comments: In section 2.2., 2.3 “Elisa testing” requires the proper reference. We added a reference (lines 211-213). Note that this is a standard immunological test. Section 2.5.1: “137-sample csv file” should be submitted in the public repository or Supplementary file. As indicated above, we need a formal authorization from the ethical committee and we are preparing a formal application for the building of public datasets. Some minor corrections to the text (style and spelling): · “audoimmunity” → “autoimmunity” The error was corrected (line 49) · “csv” → “CSV” "csv" was replaced with "CSV" on lines 248 and 255, · “Fiftly” → “Fifhtly” "Fiftly" was replaced with "Fifthly" on line 741 · “future IA alone” → “future AI alone” The french "IA" was raplaced with the (correct) "AI.Reviewer 2 Report
Comments and Suggestions for Authors
Review on “Comparison of the capacity of several machine learning tools to assist immunofluorescence-based detection of anti-neutrophil cytoplasmic antibodies” for manuscript ID ijms-2873977
I would to thank authors for the improving figures and the manuscript, but some concerns remain unclear. Whole manuscript should be carefully checked because multiple minor errors: different fonts, missing commas and stops, spaces etc. Tables should be formatted according to the journal layout.
My questions about Results and Discussion:
Unfortunately, no raw data (CSV, tabular or source images) are publicly available. I see no problem to submit training dataset and source code along with the article to reproduce the results. This study lacks novelty, simple models are used and many improvements are suggested in Discussion.
“common opinion that the success of ML is largely due to the mining of large amounts of data [49]” – that opinion paper has nothing to do with the discriminating between different fluorescence patterns.
Most of Discussion section is devoted to future steps of the study: dataset improvement, training process improvement. cell image augmentation etc.
[58] PNAS paper does not related to the cell images processing.
“Fashion-MNIST database” is questionable choice, it differs completely from the cell images used in the present study.
The overall conclusion about the manuscript – looks like a preliminary results with reasonable future stages. ML field is developing very fast, I suggest to review best practices of processing microscope images to compare with the results of this study.
Methods section comments:
In section 2.2., 2.3 “Elisa testing” requires the proper reference.
Section 2.5.1: “137-sample csv file” should be submitted in the public repository or Supplementary file.
Some minor corrections to the text (style and spelling):
· “audoimmunity” → “autoimmunity”
· “csv” → “CSV”
· “Fiftly” → “Fifhtly”
· “future IA alone” → “future AI alone”
Author Response
Answer to reviewers' comments on revised paper. “Comparison of the capacity of several machine learning tools to assist immunofluorescence-based detection of anti-neutrophil cytoplasmic antibodies” Manuscript ID :ijms-2873977 Resubmitted manuscript is provided as : - a pdf file with changes printed in red. The numbering of this file will be used in the answer to reviewers. - a docx file with a different ordering of sections, resulting in a change of citation numbering, as requested by the editor on march 1st. (reviewers' comments are printed in italic) Answer to Reviewer #1: I want to express my sincere gratitude to the authors for their comprehensive and careful revisions. The manuscript has shown considerable improvement from its initial submission, reflecting the authors' genuine commitment to resolving the previously pointed out concerns. Comments on the Quality of English Language Minor editing of English language required We thank the reviewer for his kind comment. The manuscript was read several times to remove minor errors and to fit the journal template. Answer to Reviewer #2: I would to thank authors for the improving figures and the manuscript, but some concerns remain unclear. Whole manuscript should be carefully checked because multiple minor errors: different fonts, missing commas and stops, spaces etc. Tables should be formatted according to the journal layout. We thank the reviewer for this comment. The manuscript was repeatedly checked. Indeed, in order to comply as much as possible with the amount of time allotted for revision, we performed only a minimal - admittedly insufficient - check of typing errors and grammatical correctness in the revised file submitted on februray 22. My questions about Results and Discussion: Unfortunately, no raw data (CSV, tabular or source images) are publicly available. I see no problem to submit training dataset and source code along with the article to reproduce the results. We first felt as the reviewer. We also thought this should pose no problem. Unfortunately, ethical rules became more and more severe throughout the world and particularly in France. Therefore, we asked the president of local ethics committe if we were allowed to make public fully anonymous csv files. He replied that we had to submit a formal application. Since the preparation and examination by the committee of this (admittedly quite simple) question will require at least many months, we could not include the file. However, we described our calculation methods as fully as possible, and we will be happy to provide any requested information on coding details. This study lacks novelty, simple models are used and many improvements are suggested in Discussion. This study consisted of addressing an unsolved problem: immunofluorescence image classification for ANCA diagnosis with ML, with an accuracy comparable to that provided by human readers. Novelty consisted of building a new dataset and processing it with standard ML tools. Many experimental works are performed in this manner. Indeed, many biological papers described the study of a new molecule with standard biochemical tools. Admittedly, we did not devise any novel ML method. Note that if this were the case, IJMS might not be the best suited journal to report such a theoretical advance. “common opinion that the success of ML is largely due to the mining of large amounts of data [49]” – that opinion paper has nothing to do with the discriminating between different fluorescence patterns. We think that much evidence supports the view that the effectiveness of ML is highly dependent on the content of input data as well as algorithms in many domains, including discrimination between immunofluorescence patterns. As written in reference [25] "the idea that data matters more than algorithms for complex problems was further popularized by Petr Norvig et al." (this is ref [49]). Also, this idea was suggested in a study of fluorescence classification (ref. [19] quoted in line 79). Thus, we recalled ref 19 and we added ref. 49 on line 722. Most of Discussion section is devoted to future steps of the study: dataset improvement, training process improvement. cell image augmentation etc. Indeed, we think that the discussion section of a research paper addressing a specific problem should include an analysis of the significance of obtained results and suggestions about the "road forward". The conclusion of our study is that we must first increase the dataset, then apply a number of tools, as described. [58] PNAS paper does not related to the cell images processing. This PNAS paper presented a general discussion of ML training strategy that is fully relevant to image processing : the last sentence of the Introduction reads :"in this article we present extensive measurements across image classification datasets and architectures, exposing a common empirical pattern" and the authors certainly felt that the theoretical concepts described in their study should apply to image processing. See also line 798. “Fashion-MNIST database” is questionable choice, it differs completely from the cell images used in the present study. The aim of the figure was to show that the "data hungriness" or ML is connected at the same time to qualitative and quantitative properties of samples. Therefore, we compared two fairly different sets of images of comparable size to provide a visual support to this (not very original) idea. The overall conclusion about the manuscript – looks like a preliminary results with reasonable future stages. ML field is developing very fast, I suggest to review best practices of processing microscope images to compare with the results of this study. Recent practices of processing microscope images with ML are described in depth in reference [12] and recent examples of ANA and ANCA classification were described in the introduction. The aim of our paper was rather to show that useful results could be obtained with limited datasets and fairly simple methods, which should be an incentive for the medical community to try and apply these tools to all questions they feel of interest. Indeed, many (outstanding) successes or AI are obtained with huge dat sets and highly complex models (see line 777). This might deter scientists from studying some problems they feel of interest. This was the message we wished to include. Methods section comments: In section 2.2., 2.3 “Elisa testing” requires the proper reference. We added a reference (lines 211-213). Note that this is a standard immunological test. Section 2.5.1: “137-sample csv file” should be submitted in the public repository or Supplementary file. As indicated above, we need a formal authorization from the ethical committee and we are preparing a formal application for the building of public datasets. Some minor corrections to the text (style and spelling): · “audoimmunity” → “autoimmunity” The error was corrected (line 49) · “csv” → “CSV” "csv" was replaced with "CSV" on lines 248 and 255, · “Fiftly” → “Fifhtly” "Fiftly" was replaced with "Fifthly" on line 741 · “future IA alone” → “future AI alone” The french "IA" was raplaced with the (correct) "AI.Round 3
Reviewer 2 Report
Comments and Suggestions for Authors
The authors addressed most of the comments, the situation with the raw data (CSV) is unresolvable at the moment as I can see. Authors could provide a hyperlink to some repository, where the raw data would be available as it become possible.
There are some minor errors in the main text:
Extra spaces: L64, L152, L824
Missing spaces: L636, L672
Please made usage of ROC and AUC consistent (upper case looks better, while there are acronyms).